# 4D GPR imaging of a near-terminus glacier collapse feature

Bastien Ruols[1], Johanna Klahold[1], Daniel Farinotti[2,3], James Irving[1]

[1]Institute of Earth Sciences, University of Lausanne, Lausanne, Switzerland
[2]Laboratory of Hydraulics, Hydrology and Glaciology (VAW), ETH Zürich, Zürich, Switzerland
[3]Swiss Federal Institute for Forest, Snow and Landscape Research (WSL), Sion, Switzerland

*Correspondence to*: Bastien Ruols (bastien.ruols@gmail.com)

**Abstract.** Recent advancements in drone technology now enable high-density 3D and 4D ground-penetrating radar (GPR) data acquisition over challenging glacial terrain. In this study, we present a drone-based 4D GPR dataset collected over a
surface collapse feature near the terminus of the Rhône glacier, Switzerland. The GPR measurements, repeated four times between July and October 2022, captured the evolution of an air cavity and associated subglacial drainage pathways. Our results indicate that the collapse originated where the main subglacial water channel meanders and merges with a smaller secondary channel, coinciding with a subtle step in bedrock topography. The cavity expanded progressively through a combination of subglacial melt and mechanical failure, leading to thinning of the ice roof and eventual collapse, which
manifested at the surface as circular crevasses. Downstream of the feature, the main subglacial channel underwent rapid changes in shape and size over the summer, likely driven by warm air entering from the glacier's portal and enhancing melt at the channel walls. These results highlight the capability of drone-based GPR for capturing detailed, time-dependent changes in glacier internal structure, offering new opportunities for monitoring dynamic glaciological processes in otherwise inaccessible areas.

**Short summary.** We demonstrate the use of a drone-based ground-penetrating radar (GPR) system to gather high-resolution, high-density 4D data over a near-terminus glacier collapse feature. We monitor the growth of an air cavity and the evolution of the subglacial drainage system, providing insights into the dynamics of the collapse event. This work highlights potential future applications of drone-based GPR for monitoring glaciers, in particular in regions which are inaccessible with surface-
based methods.

# 1 Introduction

Ground-Penetrating Radar (GPR) has been a core geophysical tool in the field of glaciology for over fifty years, enabling the detailed exploration of ice and bed structures across a wide variety of glaciated terrains (e.g., Woodward and Burke, 2007; Schroeder et al., 2020; Schroeder, 2022). Whereas traditional GPR acquisitions have involved collecting so-called "2D data" along one or a small number of profile lines, 3D GPR surveying, which consists of acquiring data along multiple parallel profiles, is becoming increasingly common as it offers a more comprehensive view of the subsurface. 3D GPR surveys have been conducted both from the glacier surface (e.g., Saintenoy et al., 2011; Del Gobbo et al., 2016, Egli et al., 2021a) and from helicopters (e.g., Rutishauser et al., 2016; Langhammer et al., 2018; Grab et al., 2021), offering valuable insights into ice thickness, glacier volume, and bedrock morphology (e.g., Binder et al., 2009; Saintenoy et al., 2013; Langhammer et al., 2019), as well as helping to map englacial and subglacial channels (e.g., Church et al., 2019, 2021; Egli et al., 2021a). As the field advances, two key directions have emerged: (i) increasing the spatial density of the data, and (ii) developing 4D GPR techniques, i.e., repeating the same 3D surveys over time. High-density 3D GPR involves reducing the spacing between parallel survey lines, typically to the order of the dominant GPR wavelength, to minimize spatial aliasing in the across-line direction. This enhances spatial resolution and can reveal intricate details of bed structure as well as englacial and subglacial hydrology (e.g., Murray and Booth, 2010; Reinardy et al., 2019; Church et al., 2021; Egli et al., 20221a). In contrast, 4D GPR enables monitoring of temporal changes in englacial and subglacial structures and properties, facilitating investigations of evolving thermal conditions and hydrological systems (Irvine-Fynn et al., 2006; Church et al., 2020). Despite their potential, high-density 3D and 4D GPR surveys remain rare on glaciers, largely because they require ground-based data collection, typically on foot, in often challenging or hazardous terrain. Airborne surveys by helicopter, while efficient, cannot achieve the closely spaced profile lines or positioning accuracy needed for such surveys. The latter are also prone to lateral reflections and diffractions from topography due to the large survey height above the ground surface (Forte et al., 2019).

Recent advances in drone technology have opened up promising new opportunities for GPR data acquisition that directly address these challenges (Catapano et al., 2022). In cryospheric research, drone-based GPR initially focused on small, lightweight, ultra-wideband (>1 GHz) systems for shallow (<1 m) snow depth and snow hydrology investigations (e.g., Jenssen et al., 2020, Tan et al., 2021; Valence et al., 2022). These studies successfully demonstrated the potential of drone-based GPR to derive key snow parameters, including depth, density, and liquid water content. More recently, drone-based GPR has been deployed on glaciers. Commercial systems were used by Selbesoglu et al. (2023) to compare ground- and drone-based GPR data, and by Tjoelker et al. (2024) to identify shallow buried ice within a debris-covered glacier. The development of a dedicated drone-based GPR instrument for high-resolution, high-density, 3D surveying of alpine glaciers was presented in Ruols et al. (2023). The latter study featured a large dataset comprising 462 parallel profiles spaced 1 m apart, representing over 112 line-km of data, covering an area of approximately 350 m × 500 m. The use of Real Time Kinematic (RTK) technology enables precise drone positioning and allows for the accurate repetition of flight paths, making high-density 4D GPR acquisitions over inaccessible terrains and further 4D glaciological investigations feasible.

Since the early 2000s, the frequency of glacier surface collapses near the terminus of alpine glaciers has increased significantly (Egli et al., 2021b; Hösli et al., 2025). A small number of scientific publications have focused on these events, with reports on glaciers in the European Alps (Stocker-Waldhuber et al., 2017; Kellerer-Pirklbauer and Kulmer, 2019), northern Europe (Lindström, 1993; Dewald et al., 2021), and North-America (e.g., Bartholomaus et al., 2011; Dewald et al., 2021). Compiling information from 22 Swiss glaciers, Egli et al. (2021b) suggested that such collapse events are likely driven by ice thinning and reductions in glacier ice flux directly linked to climate warming. Focusing on one event from the Otemma glacier in southwestern Switzerland, Egli et al. (2021b) hypothesized that formation begins with a meandering subglacial channel that leads to the physical removal of ice via fluvial processes. Hösli et al. (2025), focusing on one event from the Rhône glacier in central Switzerland, suggested that formation is also linked to turbulent subglacial flow because of a step in the bedrock topography which increases energy exchange between the subglacial channel and the basal ice. A large cavity is then thought to build above the unpressurized subglacial channel, either by mechanical failure from ice lamellas or by subglacial melt due to warm air entering from the water outlet at the glacier front (Egli et al., 2021b; Räss et al., 2023). This eventually leads to concentric crevasses appearing at the surface and to the collapse of the cavity at a later stage. However, in order to understand better these processes, additional information regarding the shape and evolution of the air cavity are necessary, which can be provided by high-density 4D GPR data.

In this paper, we present a high-density, high-resolution 4D GPR dataset acquired over the same surface collapse feature investigated by Hösli et al. (2025) on the Rhône glacier, which initially developed near the glacier terminus in late 2021, evolving over an approximately one-year period before eventually fully collapsing in early 2023. Between July and October 2022, we conducted four drone-based 3D GPR surveys over the evolving collapse feature using the system described in Ruols et al. (2023). Our main objective was to monitor the temporal evolution of the air cavity beneath the circular crevasses that formed at the glacier surface, along with the associated subglacial channels. Due to the size of the crevasses and the quantity of data required, such an acquisition would not be possible with classical ground-based techniques, highlighting the new opportunities provided by drone GPR. First, we present the field site and GPR data. This is followed by a detailed description of the processing workflow used to convert the raw measurements into analyzable subsurface volumes. Next, we use the data to investigate the position, size, and temporal changes of both the air cavity and the subglacial channels. Finally, we discuss these results in the context of the formation and evolution of the glacier collapse feature.

## 2. High-density 4D GPR data acquisitions

### 2.1. Field site

The Rhône glacier, located in central Switzerland (Fig. 1a), serves as a significant point of interest for glaciological research
thanks to its accessibility and historical documentation, with geodetic measurements dating back to the mid-19[th] century
(GLAMOS, 2024). Having a surface area of approximately 15.1 km$^2$ in 2021, it is the sixth largest glacier in the Swiss Alps,
even if it has experienced notable retreat in recent years (e.g., Farinotti et al., 2009; GLAMOS, 2024). The glacier flows
southwards from ~3600 to ~2200 m above mean sea level (a.m.s.l.), where it terminates in a proglacial lake that originated in
the 1990s (e.g., Tsutaki et al., 2013; Church et al., 2019; GLAMOS, 2024). Like its European neighbors, the Rhône glacier is
suffering from global warming, with a cumulative mass loss of over 15 meters of water equivalent between 2006 and 2023
(GLAMOS, 2024). The lower ablation zone of the glacier was previously investigated by Church et al. (2019, 2020, 2021)
with 2D and 3D GPR using 25-MHz antennas to characterize and monitor the englacial and subglacial drainage network. In
October 2021, large circular crevasses began to form close to the terminus of the glacier, indicating the initiation of the collapse
of an underground cavity (Hösli et al., 2025). These crevasses continued to develop throughout 2022 (Fig. 1b-c) until total
collapse occurred in early 2023 (Fig. 1d-e).

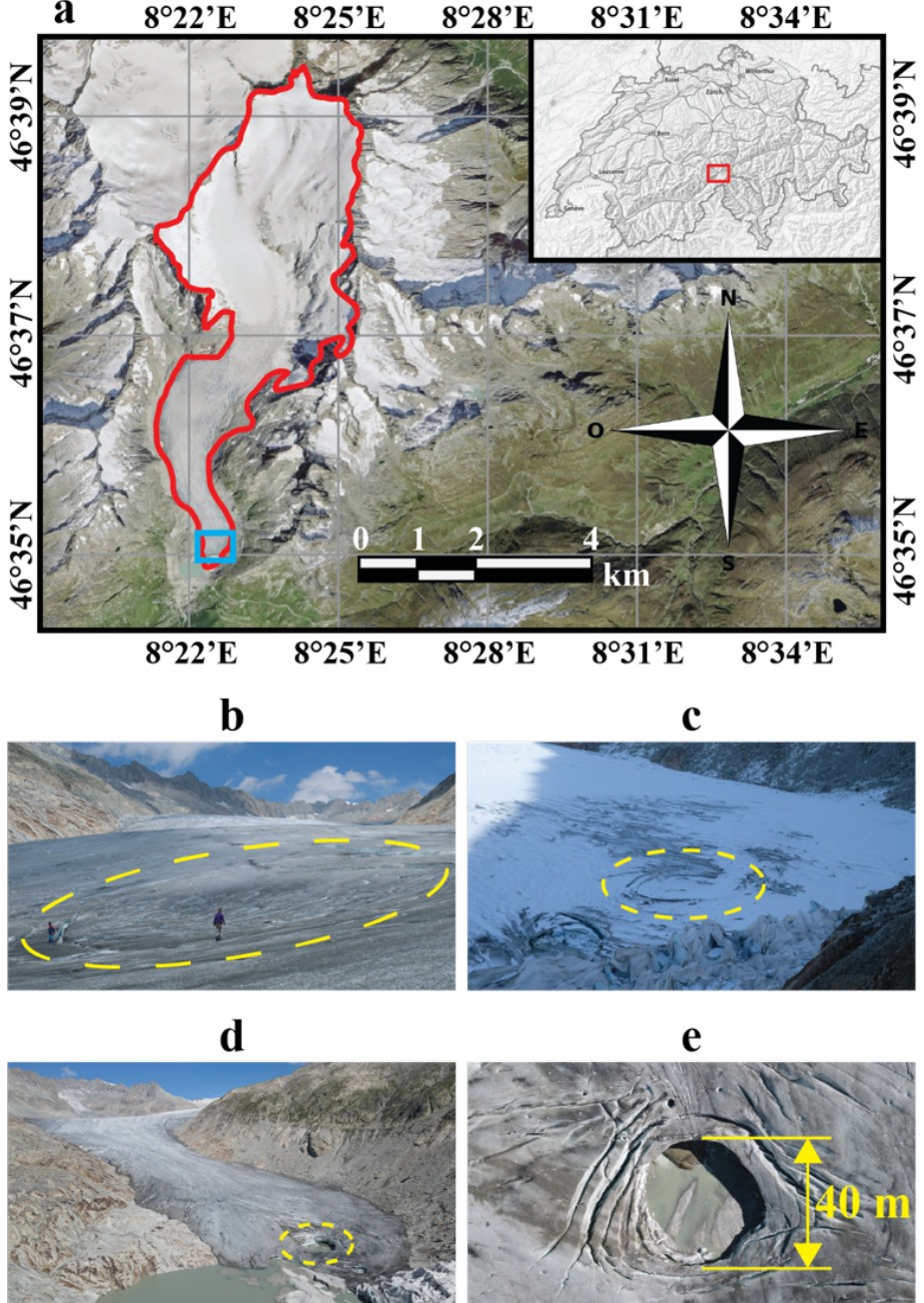

**Figure 1: Geographic localization and pictures of the field site. (a) Location of the Rhône glacier (red outline) in central Switzerland (inset). The blue square indicates the region of our GPR surveys, shown in detail in Fig. 2. Inset image from the Swiss Federal Office of Topography (Swisstopo, 2024), and background orthophotos from SWISSIMAGE 10 cm (Swissimage, 2024). (b) 28 July 2022 – collapse feature in an early stage, with a clear depression in surface elevation already visible. (c) 4 October 2022 – view of the feature from further away, at a later stage of evolution and covered by snow. (d) 11 September 2023 – view of the terminal lobe of the Rhône glacier, where the feature has now fully collapsed. (e) 11 September 2023 – nadir drone photograph of the collapsed region.**


## 2.2 Data acquisitions

Over the summer of 2022, we visited the tongue of the Rhône glacier four times to acquire high-density, high-resolution, 4D
GPR data over the evolving collapse feature, approximately once per month between July and October. The datasets were
collected using the recently developed drone-based GPR system of Ruols et al. (2023). The system comprises the following
components: (i) a DJI M300 RTK drone working with a differential GPS base station manufactured by Shenzhen DJI Sciences
and Technologies (China); (ii) a custom-designed GPR controller from Utsi Electronics Ltd (UK); (iii) a True Terrain
Following navigation system developed by SPH Engineering (Latvia); and (iv) a self-developed, ~80-MHz center-frequency,
lightweight (250-g), unshielded, resistively loaded dipole antenna acting as both transmitter and receiver. The survey
trajectories were planned with the Universal ground Control Software (UgCS), with a survey line spacing of 1 m, a target
height above the glacier surface of 5 m, and a flight speed of 4 m s$^{-1}$ (Ruols et al., 2023). To promote the most coherent
reflections from the ice-bedrock interface (Langhammer et al., 2017, 2018), parallel survey lines were flown across the glacier
perpendicular to ice flow, with the GPR antenna oriented perpendicular to the survey line direction. Figures 2a-d show the raw
trajectories of the four drone-based GPR acquisitions, and the specifications of each dataset are summarized in Table 1. The
same programmed trajectories were flown for each acquisition, with a high level of repeatability for the horizontal positioning
(Fig. 2e). However, positioning differences in elevation a.m.s.l. between the acquisitions were present due to glacier melting,
as the drone was programmed to fly at a height of 5 m above the ice surface (Fig. 2f). Note that the latter differences are
accounted for in the depth imaging of the data (Section 3.4). An illustration of the GPR system as well as a picture of it
acquiring data above the collapse feature are shown in Fig. 3. Advantages of a drone-based GPR acquisition in terms of
efficiency, safety, and practicality are clear, as high-density data could not have been acquired on the glacier surface because
of the large crevasses. Two videos from the acquisition on 4 October 2022 are provided as supplementary material: Video V1
shows the drone-based GPR system taking off and beginning the data acquisition, whereas Video V2 shows the system
acquiring data over the circular crevasses.

130          Digital OrthoPhoto (DOP) and Digital Elevation Model (DEM) data were acquired and provided by ETH Zürich's
VAW Glaciology group using a DJI Phantom 4 RTK drone. The surveys were conducted at a height of around 70 m above the
glacier surface, with ground control points (between 8 and 12 depending on the surveys) randomly scattered across the area,
for which precise locations were determined from a base station located close to the glacier. Overlapping was 70% forward,
60% sideways, and processing of the data was done using the Agisoft Metashape software.


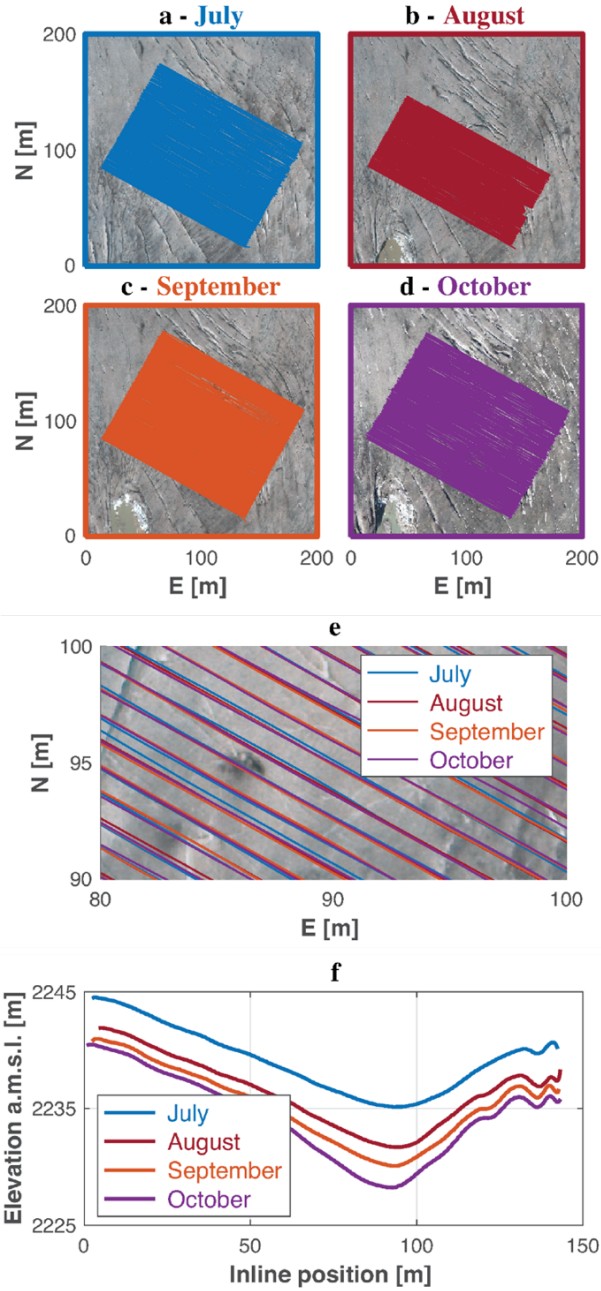

**Figure 2: Survey lines flown in the four GPR acquisitions during summer 2022. (a) to (d): Flight trajectories for the July, August, September, and October GPR acquisitions, respectively, plotted over the DOP from the same month. Northing (N) and Easting (E) are relative to 1159540 m and 2672690 m in the CH1903+ coordinate system, respectively. (e) Zoomed view of survey lines from the different acquisitions, superimposed on the same image. Note the high horizontal positioning repeatability of the drone GPR system. (f) Elevation a.m.s.l. of the drone-based GPR system along an example profile for the four acquisitions. The tracks differ in absolute elevation since the drone system is set to follow a given height above the glacier surface, which changes in time due to surface melt.**


**Table 1: GPR and DOP data specifications. Note that the August GPR dataset contains fewer profiles than the others because the drone batteries were charged to only ~66% of their total capacity for that survey.**

| | GPR | | DOP |
|---|---|---|---|
| Date | Number of profiles | Line-km of GPR data [km] | Date |
| 28 July 2022 | 104 | 14.78 | 28 July 2022 |
| 25 August 2022 | 71 | 10.02 | 24 August 2022 |
| 8 September 2022 | 108 | 15.33 | 9 September 2022 |
| 4 October 2022 | 107 | 15.32 | 7 October 2022 |


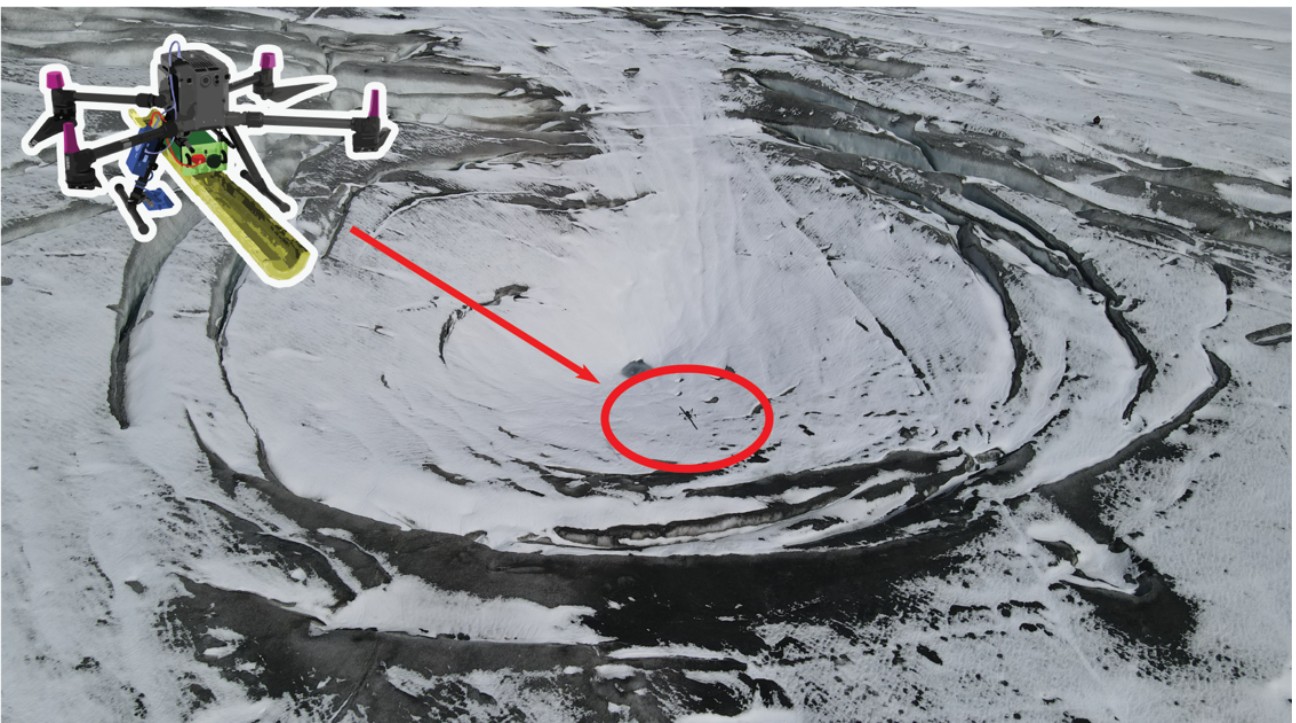

**Figure 3: The drone-based GPR system (red ellipse) acquiring data over the Rhône collapse feature on 4 October 2022.**

## 3. Data processing workflow

Our data processing workflow transforms the acquired raw GPR measurements into a 3D reflection data volume, imaged in depth, which we use to explore the internal structure of the Rhône subglacial cavity and drainage channels. This workflow involves: (i) synchronization of the drone navigation and GPR data, (ii) binning of the consecutive 3D GPR datasets onto a common grid, (iii) creation of 3D data volumes followed by basic trace processing, (iv) 3D migration, and (v) modeling of the air cavity shape along with bedrock amplitude analysis.

### 3.1 Data synchronization

In the field, two independent datasets are collected per flight: the navigation data from the drone and True Terrain Following system, and the raw GPR data from the GPR controller. The navigation data contain the information about the drone behavior, such as the flight speed, flight angles (yaw, roll, pitch), height above the ground, and precise GPS coordinates. The raw GPR data contain all the GPR traces, tagged with GPS time. In this first step, these two independent datasets are synchronized using common GPS time, with the result being that each GPR trace is associated with the corresponding navigational information (Ruols et al., 2023).

### 3.2 Binning

Next, the data from a given acquisition are segmented into individual profiles. When doing so, the positioning of each trace is corrected to take into account the distance between the GPS and GPR antennas along with the yaw, pitch, and roll angles of the drone. The latter procedure follows Ruols et al. (2023) and allows recovery of the true position of each GPR measurement. Once this is done, binning is performed to distribute the traces onto an even grid. To this end, the data positions are (i) rotated and translated into a local inline/crossline coordinate system, (ii) projected onto regularly spaced straight lines along the crossline direction having a lateral spacing of 1 m, and (iii) placed into regularly spaced bins of size 0.4 m along the inline direction. Each bin is filled with the closest GPR trace and the additional nearby traces are deleted, which Ruols et al. (2023) found to produce the highest quality GPR sections compared to trace averaging. The mean distance between the center location of each bin and the true horizontal position of the GPR trace populating that bin was found to be only 0.14 m for our data, indicating a high level of positioning precision and repeatability. It is important to note that, to effectively compare the 3D GPR datasets acquired over the summer of 2022, this binning workflow is applied to all datasets using common parameters, such that the final GPR volumes are defined on the same grid.

### 3.3 Creation of 3D data volumes

After binning, the GPR profiles from a particular acquisition are stacked side-by-side to create a 3D GPR data volume. Because the drone-based system attempts to follow the glacier surface topography while acquiring data, vertical jumps in the recording altitude can occur between adjacent traces in the crossline direction (Ruols et al. 2023). To improve the horizontal continuity

of reflection and diffraction events in the raw data volume, a static adjustment is carried out for each trace via Fourier phase shift, such that the vertical recording positions conform to a smooth acquisition surface determined by local linear regression. Doing so was found to improve the results of migration, which is discussed in Section 3.4. Further, following previous studies involving high-density 3D GPR data (e.g., Egli et al., 2021a; Ruols et al., 2023), a relative adjustment of the position of odd-versus even-numbered profiles is applied to reduce the so-called acquisition footprint effect. Specifically, because of small

internal timing delays specific to our GPR controller along with variations in the drone flight speed, a small lateral positioning adjustment between adjacent flight lines is required for best results, despite trace locations being measured with differential GPS (Ruols et al., 2023). In this regard, a constrained cross-correlation-based algorithm to determine the profile shift that maximizes the similarity between adjacent GPR profiles is applied. The results of this procedure are shown in Fig. 4 for a selected timeslice from the July 2022 dataset. After this step, the data are densified in the crossline direction using 3D linear

interpolation to double the number of parallel profiles. This reduces the directional sampling bias in the original dataset and helps to mitigate migration artifacts. The final regular grid of GPR measurements has an inline spacing of 0.4 m (from the previous binning) and a crossline spacing of 0.5 m (after interpolation). Basic GPR processing is then carried out for all data volumes, which includes: (i) mean-trace removal using a 30-trace sliding window, (ii) de-wow using a 13-point residual median filter, (iii) densification of the data in time using Fourier transform interpolation, and (iv) time-zero correction.

## 3.4 3D migration

To collapse diffraction hyperboloids in the data and to properly position reflections to their true locations in depth, we follow Egli et al. (2021a) and apply 3D topographic Kirchhoff time migration using the algorithm developed by Allroggen et al. (2014). A two-layer migration velocity model is considered, consisting of (i) an upper air layer with velocity 0.3 m ns$^{-1}$, the air layer thickness being variable and calculated from the drone altimeter data (~2 cm precision), and (ii) a lower ice layer with

velocity 0.167 m ns$^{-1}$ (e.g., Murray et al., 2000; Church et al., 2021; Egli et al., 2021a). Although the programmed drone flight height above the glacier surface was set to 5 m, this value varies during acquisition due to the flight velocity and the drone's attempt to follow the glacier surface topography in real-time using the TTF system. Considering a constant radar velocity for glacier ice is a standard procedure for both ground-based and airborne GPR surveys (e.g., Langhammer et al., 2017; Grab et al., 2021; Church et al., 2020), even if the effects of internal heterogeneities like water- or air-filled features are neglected. A

migration aperture of 40 m was found to produce the most satisfactory results and is consistent with the maximum width of diffraction hyperboloids observed in the Rhône datasets. After migration, the vertical time axis is converted to depth using the assumed velocity model, which yields the final GPR data volume in space. Note that, in applying this 3D migration procedure, we inherently assume a uniform antenna radiation pattern. Although we acknowledge that this is not entirely correct for an unshielded dipole antenna, we see little indication in our data of a strong radiation pattern footprint. Indeed, diffraction circles

in time slices show little variation in intensity between the in-line and cross-line directions, and diffraction hyperboloids present in the unmigrated data are observed to correctly focus upon migration.

Figures 5a-c show part of the inline profile at a crossline position of 76 m for the July 2022 acquisition, plotted in

time before migration, in time after migration, and in elevation a.m.s.l. after migration, respectively. The profile shows three main features: (i) a continuous bedrock reflection, (ii) a feature that we interpret to be the roof of the subglacial cavity, and (iii) another feature that we interpret to be a multiple bedrock reflection from within the cavity. The interpretation of the cavity roof is supported by reflections in the neighboring profiles, which show a consistent pattern. Figure 6 displays the migrated and depth-converted data in 3D through three selected inline profiles, crossline profiles, and depth slices. Note that the crossline profiles and depth slices represent planes extracted from the 3D data volume that was built from the acquired and interpolated parallel inline profiles.

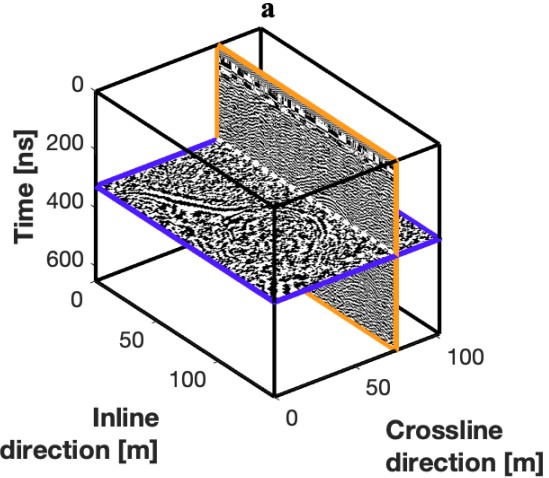

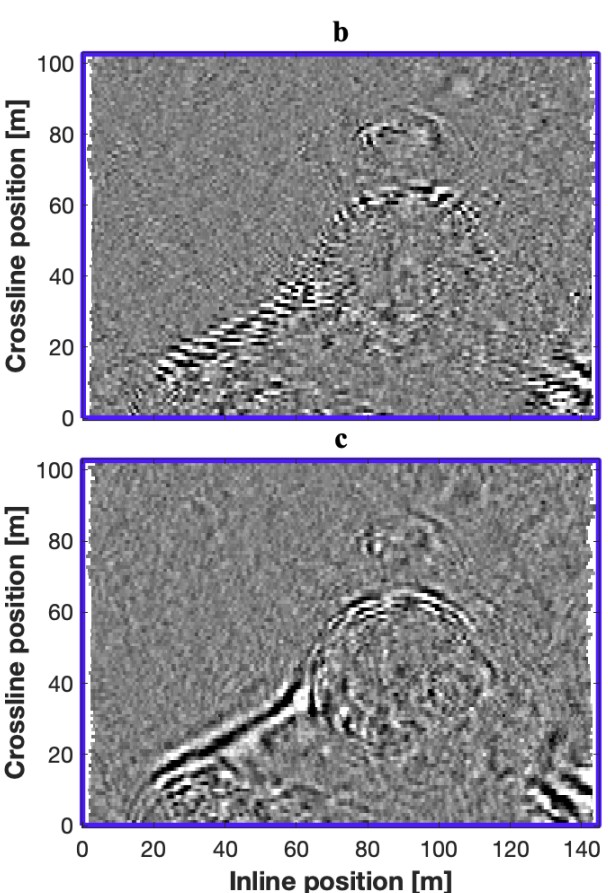


**Figure 4: Results of applying cross-correlation-based profile shifting to the July 2022 dataset. (a) 3D view of the data volume highlighting the timeslice at 325 ns (purple) displayed in (b) and (c), as well as the inline profile at a crossline position of 76 m (orange) displayed in Fig. 5. (b) and (c): Timeslice at 325 ns before and after application of the shifting, respectively.**

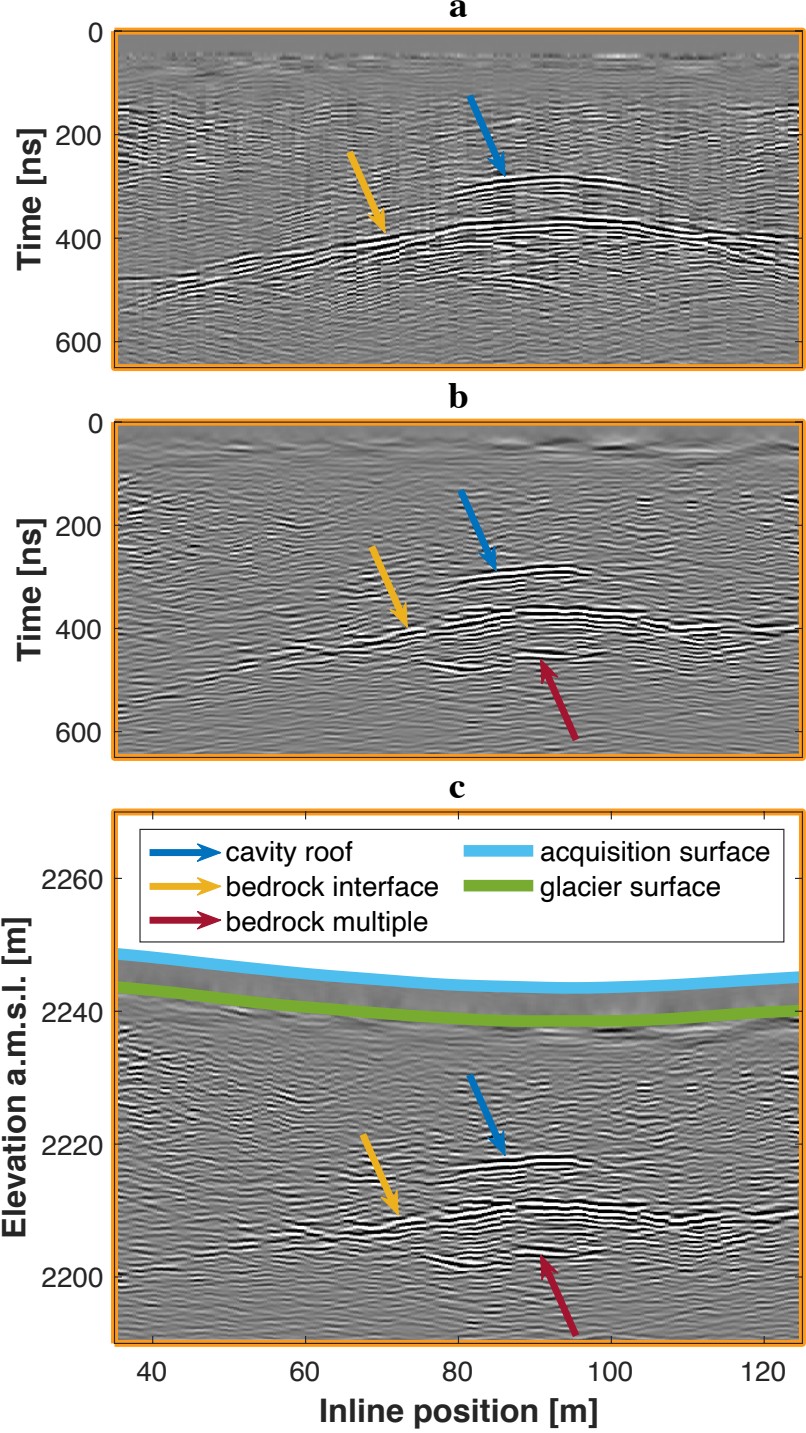

Figure 5: Results of applying 3D migration to the July 2022 dataset, showing part of the inline profile at a crossline position of 76 m (Fig. 4a) (a) in time before migration, (b) in time after migration, and (c) in depth after migration.

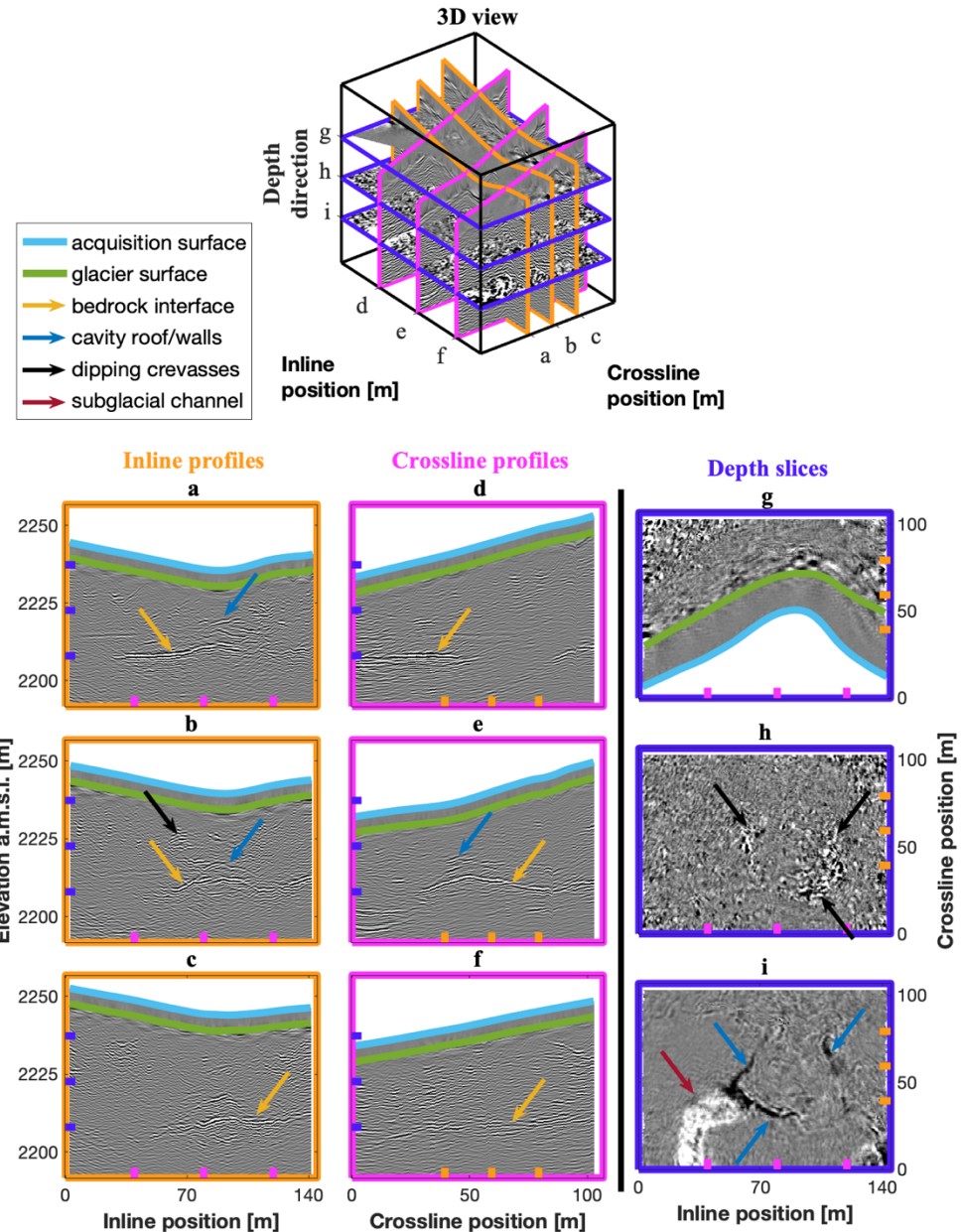

**Figure 6: 3D visualization of the July 2022 dataset. The 3D view (top) shows the location of the three selected inline profiles (orange), crossline profiles (pink), and depth slices (purple). (a) to (c): Inline profiles at crossline positions of 39 m, 59 m, and 79 m, respectively. (d) to (f): Crossline profiles at inline positions of 40 m, 80 m, and 120 m, respectively. (g) to (i): Depth slices at 2238 m, 2224 m, and 2208 m a.m.s.l., respectively.**

## 3.5 Bedrock surface, air cavity modeling, and amplitude analysis

We construct a 3D model of the bedrock reflection surface for each dataset. To this end, bedrock picking is performed manually on the inline profiles (Fig. 7a), the results of which are verified on both the crossline profiles and depth slices, with visible
inconsistencies being removed. Next, we fit a preliminary smooth surface, determined by local linear regression, to these picks (Fig. 7b). After removing obvious outliers, a new surface is fitted to the picks to yield the final estimate of the bedrock surface (Fig. 7c). It is important to note that an apparent circular rise in this surface is observed at the location of the collapse feature. This is a migration artifact that results from the assumption of a constant radar wave speed in the glacier ice, when in fact an air-filled cavity exists at this location. However, this bedrock "pull-up" artifact does not impact the bedrock reflection
amplitude analysis described below.

Similar to what is done for the bedrock surface, the roof of the air cavity is also modeled by picking the ice-air interface in each dataset and fitting a smooth surface to these results. In order to investigate the evolution of the cavity, we compute the minimum ice roof thickness and the maximum air cavity height by considering the distances between the roof of the air cavity and (i) the glacier surface, and (ii) the estimated bedrock surface, respectively. The corresponding results are
presented in Table 2. Regarding the maximum cavity height, note that these estimates have been corrected for the presence of air within the cavity.

A commonly used technique to identify the position of subglacial channels in 3D GPR data is to analyze the amplitude characteristics of the reflection near the glacier bed (Egli et al., 2021a; Church et al., 2021). Because of the stronger contrast in dielectric permittivity between ice and water or air than between ice and bedrock, subglacial channels are expected to be
associated with higher GPR reflection amplitudes (e.g., Wilson et al., 2014; Church et al., 2019; Egli et al., 2021a). To extract amplitude information along the glacier bed, we follow Egli et al. (2021a) and apply a linear Fourier phase shift to each trace to flatten the data along the estimated bedrock surface, which is followed by calculation of the instantaneous amplitude attribute using the Hilbert transform (e.g., Taner et al., 1979; Chopra and Marfurt, 2007). To compensate for any errors in the estimated bedrock location, the maximum reflection strength is computed for each individual trace over a vertical 2-m window containing
the bed reflection (Egli et al., 2021a). Traces located less than 5 m from the border of the GPR grid are not considered to avoid boundary effects related to suboptimal 3D migration. The results of the bedrock reflection amplitude analysis for the four GPR acquisitions, superimposed over the corresponding DOPs, are displayed in Fig. 8.

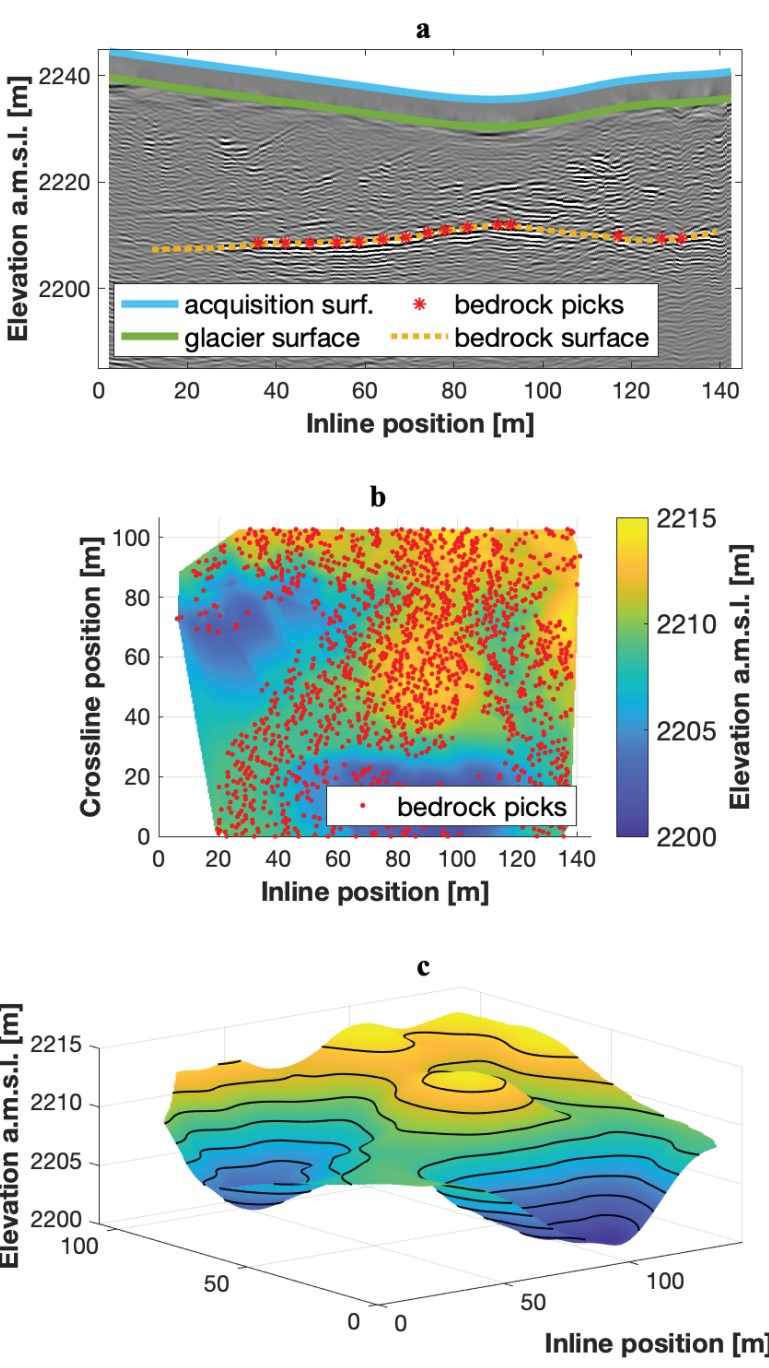

**Figure 7: 3D model of the bedrock surface for the July 2022 dataset. (a) Example inline profile from Fig. 6a with bedrock pick locations (red) and corresponding smoothed surface (yellow). (b) Locations of the picks for the whole dataset, plotted over the preliminary smoothed bedrock surface. (c) 3D view of the final bedrock surface model.**


**Table 2: Air cavity characteristics derived from the 4D GPR measurements.**

| Dataset | Minimum ice roof thickness [m] | Maximum cavity height [m] |
|---|---|---|
| July | 9.6 | 15.9 |
| August | 6.1 | 16.8 |
| September | 5.3 | 16.9 |
| October | 3.0 | 18.4 |

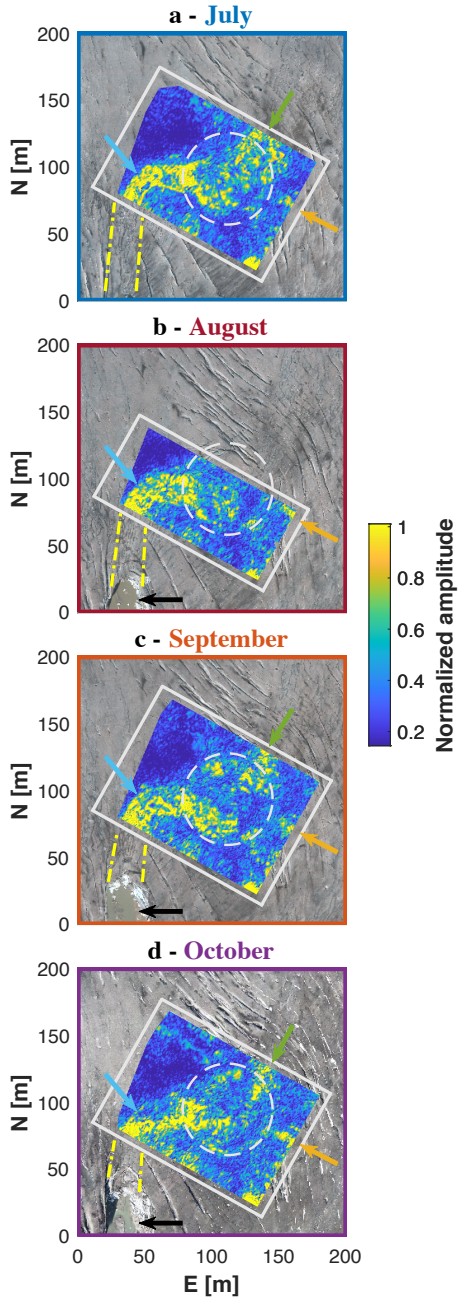

**Figure 8: Results of the amplitude analysis at the bedrock interface for the four acquisitions, plotted over the DOP from the same month. The white squares depict the borders of the corresponding GPR dataset. The dashed white circles indicate the approximate location of circular crevasses. The green and yellow arrows show the potential locations of two subglacial channels entering the collapse feature from the northeast and southeast, respectively (see Fig. 11). The blue arrows indicate the subglacial channel leaving the feature westwards and turning towards the south. The dashed yellow lines highlight the potential connection with the channel outlet (black arrows) visible on the DOP. Northing (N) and Easting (E) are relative to 1159540 m and 2672690 m in the CH1903+ coordinate system, respectively.**

## 4. Results

The 3D visualization of the July 2022 dataset presented in Figure 6 clearly shows a number of glacier internal structures. In the inline and crossline profiles in Figures 6a-f, strong reflections related to the bedrock interface (yellow arrows), the air cavity roof (blue arrows), and dipping crevasses (black arrow) can be observed. In the depth slice presented in Figure 6h, we see large circular reflections from concentric crevasses (black arrows) which can be viewed in greater detail and context in video supplement V5. The depth slice in Figure 6i reveals the walls of the circular air cavity (blue arrows), and a suspected subglacial channel leaving the collapse feature (red arrow), which can also be observed in the results of our amplitude analysis discussed below. For further visualizations, videos V3 to V14 contain animations providing insights into the 3D models obtained for all four surveys.

In the bedrock amplitude analysis results presented in Figure 8, we can observe the potential locations of two subglacial channels entering the collapse feature, corresponding with regions of higher reflection strength: (i) a main channel entering from the northeast, and (ii) a smaller channel entering from the southeast. Within the collapse feature, the bedrock reflection amplitude is heterogeneous, but the trend of a meander turning from the northeast to the west can be seen. Further down glacier, the suspected subglacial channel leaving the collapse feature westwards appears to turn towards the south, in the direction of the channel outlet visible on the background DOP. Note that there is another high amplitude anomaly, located in the lower right corner of each subfigure, which may represent a separate water body or cavity, but lack of additional data prevents us from drawing any further conclusions.

Figure 9 shows the evolution of parts of two selected inline profiles, whose positions are indicated in Fig. 9a. The first profile (Fig. 9b-e) spans the location where the main subglacial channel leaves the collapse feature and shows a distinct internal reflector which we interpret to be the channel roof. This reflector is not yet separable from the bed reflection in the July survey (Fig. 9b), but it becomes apparent in August (Fig. 9c) and evolves over time (Fig. 9d-e). In the October survey (Fig. 9e), a second reflection becomes visible above the subglacial channel, which is possibly related to the appearance of fractures caused by subsidence of the channel roof. The second considered inline GPR profile (Fig. 9f-i) passes through the center of the main collapse feature. Here, a continuous reflector remains visible for all surveys, which we interpret as the roof of the underlying air cavity. Note that, at this location, the glacier surface elevation can be seen to decrease throughout the summer. This is a result of both surface ice melt and subsidence, the latter possibly being due to a combination of ice creep into the cavity and partial mechanical failure.

Finally, Figure 10 displays the resulting 3D models of the air cavity for the July, August, September, and October acquisitions, along with the estimated bedrock and glacier surfaces for reference. Tracking the evolution of the cavity over the summer of 2022, it appears that the ice roof becomes thinner while the height of the cavity increases (Table 2). From July to September, the shape of the cavity remains rather constant, whereas it appears to widen in October.

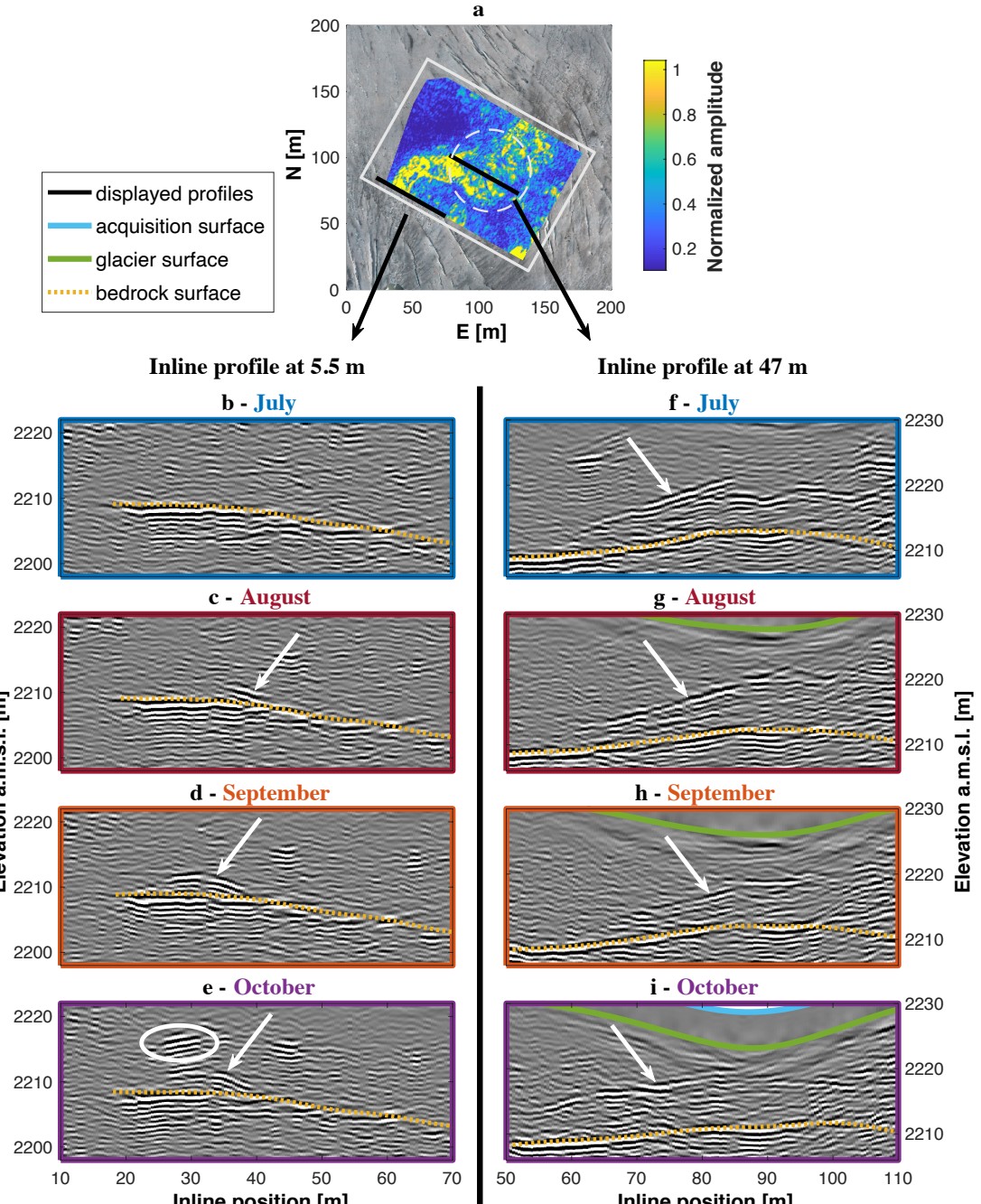

**Figure 9: Visualization of the Rhône glacier datasets over the summer of 2022 through a comparison of two inline profiles. (a) Amplitude analysis result for the July dataset, plotted over the DOP from the same month. The white square depicts the border of the GPR grid. The two black lines show the sections of the two selected profiles, located at crossline positions of 5.5 m and 47.0 m. Northing (N) and Easting (E) are relative to 1159540 m and 2672690 m in the CH1903+ coordinate system, respectively. (b) to (e):**
**Inline profiles at 5.5 m, focusing on the channel outlet. The white arrows and circle indicate the channel roof and a large englacial reflection, respectively. (f) to (i): Inline profiles at 47.0 m, focusing on the collapse feature. The white arrows indicate the roof of the underlying air cavity.**

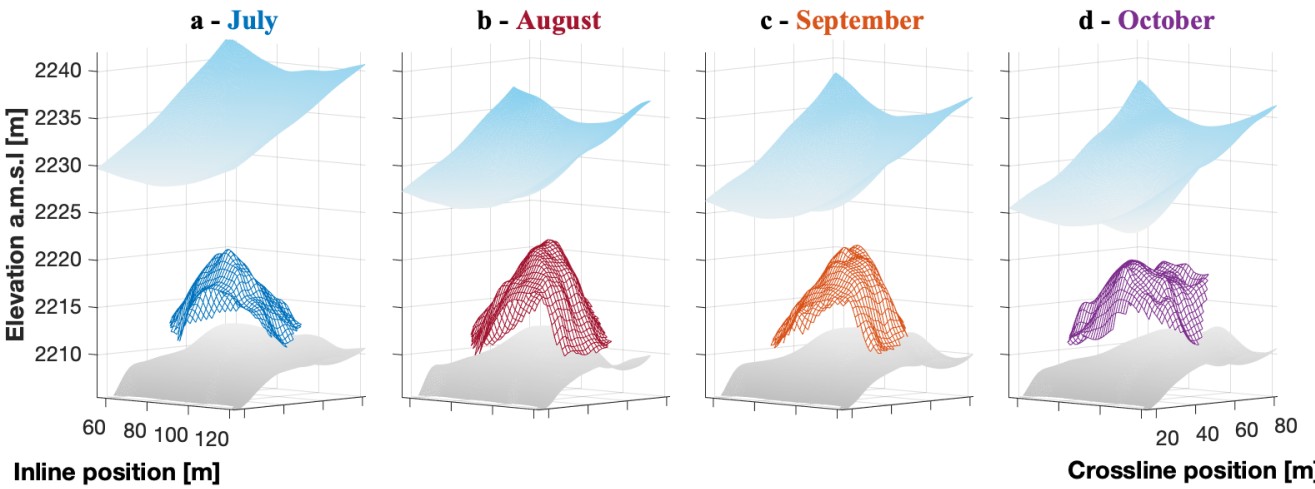

**Figure 10: Evolution of the collapse feature deduced from the GPR measurements. (a) to (d) show the 3D models of the underlying air cavity over the summer of 2022, with the bedrock surface plotted in light grey and the glacier surface plotted in light blue for reference.**

## 5. Discussion

The results of the amplitude analysis (Fig. 8) suggest that the Rhône collapse feature formed at a particular location near the glacier snout where (i) the main subglacial channel forms a meander, and (ii) a secondary channel enters the main channel. Examination of a photograph taken in the summer of 2023 after full collapse of the cavity (Fig. 11) also indicates a localized step (~1 m in height) in the bedrock topography, which was further confirmed in Hösli et al. (2025). The combination of these three factors could have led to water turbulence and related energy dissipation, which could ultimately be the process by which the cavity was initiated. In contrast, although sediments are visible at both the glacier terminus and inside of the collapsed feature (Figure 11), we do not have any evidence that would suggest that these sediments played a significant role during the formation of the feature itself. This is dissimilar to the situation described by Stocker-Waldhuber et al. (2017), who analyzed the formation of a surface collapse feature for Gepatschferner, Austria, and reported that the subsidence of the glacier surface accelerated significantly after a heavy precipitation event evacuated a sediment layer of more than 10-m thickness.

For what the relative magnitude of the two subglacial channels is concerned, the qualitative interpretation of glacier-wide DEMs available for both the glacier surface and subglacial topography (data from Swisstopo (2024) and Grab et al. (2021), respectively) as well as expert judgement based on in-situ perceptions (e.g. the "loudness" of water-related sounds that can be heard emerging from the glacier interior), suggest that the channel originating from the northeast is likely to drain the majority of the glacier's subglacial water system, whereas the second one, originating from the southeast, likely drains a constrained hydrological basin on the orographic left-hand side of the glacier. Downstream of the collapse feature, a single large subglacial channel is seen to be flowing towards the west and then turning towards the south, i.e. in the direction of Rhône glacier's proglacial lake. Our interpretation of the channel pathways, which is primarily based on the amplitude analysis of the bedrock reflection, was later confirmed by visual observation in September 2023, i.e. after the full collapse of the subglacial cavity (Fig. 11). This ground truthing is amongst the most compelling evidence for the potential that lies in the applied drone-based survey methodologies.

Further down glacier, the main channel outlet is also seen to be evolving over time (Fig. 8): it has a meandering yet comparatively narrow shape in July, slowly straightens and enlarges during summer, and appears almost straight in October. A distinct reflection from what we interpret as the channel roof appears in August (Fig. 9c) and rises from the bedrock over the course of the summer (Fig. 9d-e). Above this channel roof, further internal reflections appear in October, which could stem from air-filled fractures that form as ice lamellas start detaching and falling into the channel. The imaged part of the channel is located close to the glacier portal which consists of an ice arch measuring approximately 10 m in height (Fig. 12), and from August the channel is likely unpressurized and air-filled. In such conditions, warm air can enter the channel from the glacier portal, and we suggest that the heat advected in this way may have contributed to ice melt at the channel walls, thus favoring the fast evolution seen with our 4D imaging.

Finally, the temporal evolution of the main cavity beneath the circular crevasses could also be monitored with our 4D survey (Fig. 9f-i and Fig. 10). The results show that the thickness of the ice roof decreases while the height of the cavity

increases as the collapse feature evolves throughout the summer (Table 2). The shape of the cavity remains similar between July and September, while it appears to have increased in width in October (Fig. 10). Egli et al. (2021b) suggested that warm air can enter cavities from the glacier front when a widely open and unpressurized channel connects the glacier portal to a cavity. In our case, however, the channel between the portal and the cavity does not seem air-filled until August (Fig. 9b-e), i.e., well after the cavity started to grow. This makes air circulation an unlikely driving mechanism up to that stage. After August, when the subglacial channel appears to be air-filled, this process may have played a role and enhanced the air cavity evolution. As an alternative, Räss et al. (2023) hypothesized that the collapse feature at Rhône glacier grew by mechanical failure of ice lamellas and the subsequent melting and transport of the ice by the subglacial stream – a process referred to as "block caving" in Paige (1956) or Egli et al. (2021b). This latter hypothesis is supported by Figure 13, showing two images from inside the air cavity acquired in August 2022. Large blocks of ice can be seen resting on the cavity floor, which must have collapsed from the ice roof and are likely to deplete over time through a combination of melt and fluvial transport. The latter observation also explains the heterogeneous results in amplitude analysis at the bedrock interface beneath the collapse feature (Fig. 8). The interplay between the various processes causes the question of which of them might dominate the overall evolution.

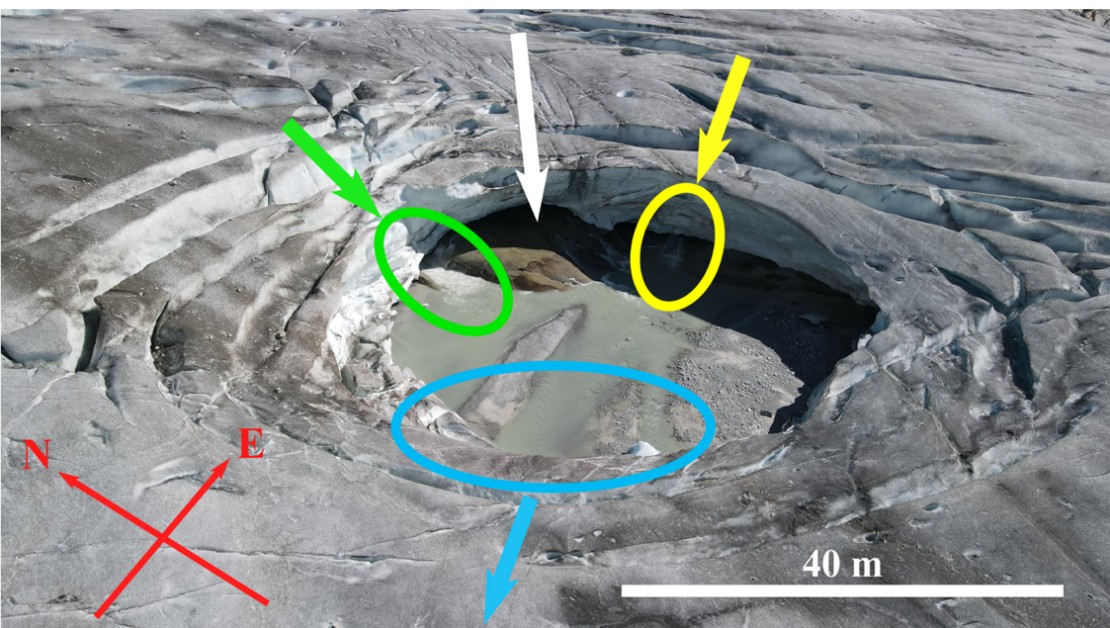

**Figure 11: Picture of the studied feature after its full collapse, taken on 11 September 2023. Two water channels entering the feature from the north (green) and east (yellow) are visible, as well as one leaving the feature towards the west (blue). The white arrow points to an observed step (~1 m in height) in the bedrock topography.**

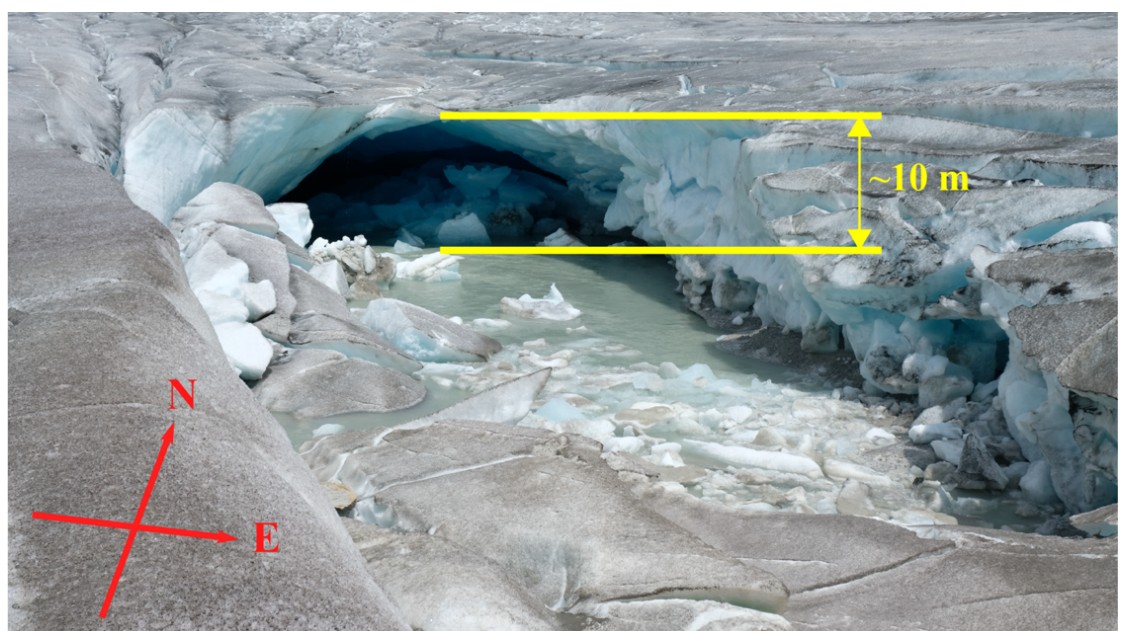

**Figure 12: Picture of the channel outlet at the glacier front, taken on 28 July 2022.**

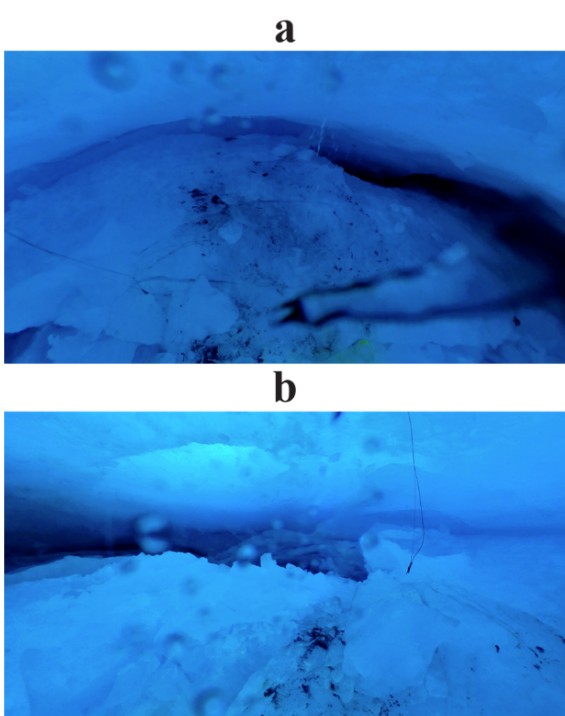

375 **Figure 13: Images taken from a video recorded inside the subglacial cavity by the ETH Zürich VAW Glaciology group. (a) Towards the east. (b) Towards the west. Large blocks of ice that have fallen from the cavity roof are seen to have accumulated on the bedrock surface.**

## 6. Conclusion

In this paper, we presented the analysis and interpretation of a high-density 4D GPR dataset collected over an evolving surface collapse feature near the terminus of Rhône glacier (Switzerland). Our study was made possible by a newly developed drone-based GPR system which allows for the acquisition of high-quality, precise, repeat measurements. While the surveyed area (approximately 100 m x 150 m) might be considered to be relatively small, our results show the strong potential for using repeated, high-resolution GPR surveys in combination with amplitude analysis for investigating the detailed temporal evolution of the subglacial drainage system. We anticipate that the demonstration of these new capabilities will spur a new line of investigations, ultimately resulting in important glaciological advances.

The GPR data acquired in this study allow two main conclusions to be made. The first relates to the genesis of the collapse feature: it initially emerged at a location where the main subglacial water channel meanders and merges with a secondary smaller water channel, and where the bedrock topography displays a small step. After initiation, the subglacial cavity grew by a combination of processes including subglacial ice melt and mechanical failure, with lamellas of ice detaching from the cavity roof. This led to a continuous decrease of the ice roof thickness, favoring further instability. At the surface, these processes resulted in a set of visually concentric circular crevasses, ultimately resulting in the mechanical failure of the cavity roof. The second conclusion relates to the temporal evolution of the main subglacial channel downstream of the cavity. Here, the channel was seen to quickly evolve during summer, both in shape and in size. Since the channel appears to have been unpressurized for most of the summer, we attribute this fast evolution to the advection of warm air from the large portal observed at the glacier front, and to the accelerated melt happening at the channel walls through this advection.

On a broader perspective, and whilst acknowledging that an investigation focusing on an individual collapse feature cannot be used to establish a potential link to ongoing climate change, our study provides specific, complementary information to larger-scale studies that have provided evidence for an increase in the frequency by which surface collapse features occur (Egli et al., 2021b; Hösli et al., 2025). In combination, this growing body of literature sheds light on a phenomenon that has sparked curiosity in the recent past, and clearly associates it with the ongoing process of glacier thinning and related reduction in ice-flow velocities – two processes that are clearly driven by climate change and rising temperatures in particular (e.g., Hugonnet et al., 2021; Troilo et al., 2024; The GlaMBIE Team, 2025). As we expect that glacier surface-collapse features will emerge in other parts of the world too as glaciers continue to thin, our study contributes to better understand the local-scale processes and effects that such features have.

**Video supplements.** They will be uploaded and doi provided on a dedicated platform as requested in due time after review. For now, they are available at the following url with the password 'collapse_22':

https://unils-my.sharepoint.com/:f:/g/personal/bastien_ruols_unil_ch/Ekwcougm41FCvXsj0LYSn7gBc2dNsUUyrgkGPlkG4CZl3A?e=J8PIuN

- **Video V1: Drone-based GPR system taking off and beginning the data acquisition on 10 October 2022.**
- **Video V2: Drone-based GPR system acquiring data over the circular crevasses on 10 October 2022.**
- **Videos V3-V14: Animations showing the full 3D models obtained for the four surveys.**

**Supplementary Material.** One .pdf file was uploaded alongside this manuscript.

**Author contributions.** Bastien Ruols and Johanna Klahold planned and conducted fieldwork together. Bastien Ruols designed the GPR surveys and operated the drone-based GPR system. Daniel Farinotti gave access to the DOPs used in this study. Bastien Ruols processed and interpreted the drone-based GPR data under the supervision of James Irving. Bastien Ruols wrote the manuscript, which was revised by James Irving and Daniel Farinotti. For analysis and discussion, James Irving and Daniel
Farinotti provided geophysics and glaciology expertise, respectively. The final version of the manuscript was reviewed by all co-authors.

**Competing interests.** Some authors are members of the editorial board of *The Cryosphere*.

**Acknowledgments.** We thank Mélissa Francey, Pascal Egli, Ben Robson, and Alexi Morin for their substantial help with fieldwork on the Rhône glacier, as well as Gabriela Clara Racz for her insights into the initial manuscript. We also thank Margot Sirdey and Flavio Calvo for their help with implementing the 3D migration code in parallel, as well as Huw Horgan for reading the initial manuscript and providing comments. We acknowledge the Laboratory of Hydraulics, Hydrology and Glaciology (VAW) for synchronizing fieldwork, providing the photogrammetry data, and sharing some findings about the
Rhône collapse feature, in particular Christophe Ogier, Elias Hodel, and Leo Hösli. We acknowledge the use of OpenAI's ChatGPT language model for assistance with editing and refining the text of this manuscript (OpenAI, 2024). Finally, we thank the two anonymous reviewers as well as editor Kristin Poinar for comments that greatly helped to improve the quality of the manuscript. This work was supported by grants from the Swiss National Science Foundation to J. Irving (grant number 200021_188575) and D. Farinotti (grant number 200021_212061).

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
