# Peer review of "4D GPR imaging of a near-terminus glacier collapse feature"

_EGUsphere, 2024_

## Author Response (AR1)

Dear Professor Poinar,

First, please note that we modified the title of the manuscript to make it more concise and direct: "4D GPR imaging of a near-terminus glacier collapse feature".

We greatly appreciate the time that you and the reviewers have taken to examine our manuscript. We found the insights from both reviewers to be very helpful and have made efforts to address each point in our updated manuscript. We also would like to thank you for your availability and understanding when it came to extend the resubmission deadline. All the requested material has now been submitted.

Please note that video supplements are still only available on a temporary platform, but that they will be uploaded on a dedicated online platform in due time after the revision process. Also, please note that the Supplementary Material file hasn't been updated since no reviews were done on it, the document provided at the initial submission is therefore still accurate.

We again would like to thank you and hope that the updated version meets the publication standards of The Cryosphere.

Best regards,

Bastien Ruols, on behalf of all co-authors

**Reviewer 1:**

We sincerely thank Reviewer 1 for their thoughtful and constructive feedback. The suggestions regarding the structure significantly improved the readability and flow of the manuscript, and the remarks about situating our work within a broader context helps to better frame our contribution. We are grateful for the time and care taken in the review, which has contributed meaningfully to the improvement of our work.

**Abstract**

I suggest slightly reducing the detailed results in the abstract and focusing on a concise summary of the findings. This would better emphasize the innovation of the applied methodology as a key tool, enabling new insights into glaciological processes, strengthening the connection between the method and the results.

Thank you for this suggestion. The revised abstract now reads:

*Abstract: "Recent advancements in drone technology now enable high-density 3D and 4D ground-penetrating radar (GPR) data acquisition over challenging glacial terrain. In this study, we present a drone-based 4D GPR dataset collected over a surface collapse feature near the terminus of the Rhône glacier, Switzerland. The GPR measurements, repeated four times between July and October 2022, captured the evolution of an air cavity and associated subglacial drainage pathways. Our results indicate that the collapse originated where the main subglacial water channel meanders and merges with a smaller secondary channel, coinciding with a subtle step in bedrock topography. The cavity expanded progressively through a combination of subglacial melt and mechanical failure, leading to thinning of the ice roof and eventual collapse, which manifested at the surface as circular crevasses. Downstream of the feature, the main subglacial channel underwent rapid changes in shape and size over the summer, likely driven by warm air entering from the glacier's portal and enhancing melt at the channel walls. These results highlight the capability of drone-based GPR for capturing detailed, time-dependent changes in glacier internal structure, offering new opportunities for monitoring dynamic glaciological processes in otherwise inaccessible areas."*

**Introduction**

The introduction effectively sets the stage, but a broader explanation of the advantages and limits of 4D drone/helicopter-based GPR compared to other ground-based methods would strengthen the impact. For example, discussing how non-ground-based systems improve safety, enable data collection in inaccessible or hazardous areas, and allow for high-resolution, repeated surveys over short timescales could highlight their importance for advancing glaciological research. At the same time, acknowledging potential limitations, such as challenges in vertical positioning accuracy due to glacier surface changes or lateral obstacles as well as the interaction between the 3D GPR signal lobes with the topography (e.g. see Forte et al., 2019), would provide a balanced perspective and enhance the methodological transparency of the paper.

We thank the reviewer for this comment and have tried to better clarify these individual points throughout the first two paragraphs of the introduction (Section 1). Where we partially disagree,

is with the comment related to the challenges related to vertical positioning accuracy, as we do not believe this is a significant issue with our drone-based system. While the vertical position of the drone varies between acquisitions and along each survey line, it is measured precisely with RTK technology, and this variation, along with the drone height above the glacier surface, are accounted for when migrating the datasets and producing the depth images. As for the interaction between the lobes of the GPR antenna radiation pattern and topography, please see our response to the same comment under Methods below.

L 51: Consider adding "and in proximity of strong lateral reflectors (Forte et al. 2019)."

Thank you for this suggestion. We have added the following sentence to the first paragraph of the introduction. The revised text now reads:

> *Section 1: "The latter are also prone to lateral reflections and diffractions from topography due to the large survey height above the ground surface (Forte et al., 2019)."*

L 54-61: There is a slight imbalance in the details provided for Ruols et al. (2023) compared to the other works (Jenssen et al., 2020; Tan et al., 2021; Valence et al., 2022). Rephrasing this section and including recent works such as Tjoelker et al. (2024) and SelbesoĞlu et al. (2023) would create a more balanced discussion.

Thank you for pointing out this imbalance. The revised text now reads:

> *Section 1: "These studies successfully demonstrated the potential of drone-based GPR to derive key snow parameters, including depth, density, and liquid water content. More recently, drone-based GPR has been deployed on glaciers. Commercial systems were used by Selbesoglu et al. (2023) to compare ground- and drone-based GPR data, and by Tjoelker et al. (2024) to identify shallow buried ice within a debris-covered glacier."*

L 92: Please, be more specific about "recently" as lakes formed during 1990s and in 2005 according to Tsutaki et al. (2013). I also suggest providing some more context concerning glacier changes (including collapse) in relation to the ongoing climate change.

Thank you for pointing this out. We have revised the text to clarify that the lakes formed in the 1990s (the two different lakes discussed in Tsutaki et al. (2013) have merged since then). We also added a sentence about the cumulative mass loss of the Rhône glacier between 2006 and 2013 to highlight the impact of global warming. The revised text now reads:

> *Section 2.1: "The glacier flows southwards from ~3600 to ~2200 m above mean sea level (a.m.s.l.), where it terminates in a proglacial lake that originated in the 1990s (e.g., Tsutaki et al., 2013; Church et al., 2019; GLAMOS, 2024). Like its European neighbors, the Rhône glacier is suffering from global warming, with a cumulative mass loss of over 15 meters of water equivalent between 2006 and 2023 (GLAMOS, 2024)."*

**Methods (acquisition and processing)**

While the authors often refer to a previous paper (Ruols et al. 2023) for methodological details, adding a few more information directly in this manuscript would improve clarity and accessibility. Unless I missed it, I suggest adding some information about:

• The drone set-up:
      o L 112: Is it a shielded antenna?
      o A zoomed picture of the drone-GPR system could be added as a box to Fig. 3

The antenna is an unshielded, resistively loaded dipole antenna, to minimize weight as much as possible (Ruols et al. 2023). The word 'unshielded' was added to the antenna description:

*Section 2.2: "...(iv) a self-developed, ~80-MHz center-frequency, lightweight (250-g), unshielded, resistively loaded dipole antenna acting as both transmitter and receiver."*

An illustration of the drone was added to Figure 3. Thank you for this comment which greatly improve the value of this figure, which is now introduce in the text as:

*Section 2.2: "An illustration of the GPR system as well as a picture of it acquiring data above the collapse feature are shown in Fig. 3."*

• the development of the flight plan:
      o I did not understand if/how cross-profile were acquired or interpolated, as in Fig. 2 the acquisition seems to be done along parallel profiles only, but in Fig. 6 both inline and crossline profile are shown).

Indeed, the cross-line profiles were not flown, but were rather extracted from the 3D data volume comprised of interpolated parallel in-line profiles. To address this point, we have added the final sentence to the text when we initially introduce Figure 6 (Section 3.4):

*Section 3.4: "Note that the crossline profiles and depth slices represent planes extracted from the 3D data volume that was built from the acquired and interpolated parallel inline profiles".*

• How the 3D GPR signal lobes interact with the topography in particular in relation with the height above the ground and the angle between the GPR and the surface?

All processing of our GPR data (most notably the topographic Kirchhoff migration procedure) assumes a uniform antenna radiation pattern. Although we acknowledge that this is not entirely correct for an unshielded dipole antenna, we see very little indication in our data of a strong radiation pattern footprint. Indeed, diffraction circles in time slices show little variation in intensity between the in-line and cross-line directions, and diffraction hyperboloids present in the unmigrated data correctly focus upon migration. The uneven topography of the ice surface is also considered in our migration velocity model, which includes the air layer of variable thickness between the drone and the glacier. We also note that in Ruols et al. (2023), we conducted various tests to assess the impact of height above the surface on the GPR data acquired by our system. These tests showed that no significant loss of information regarding the internal structures of the ice occurs when flying between 1 and 10 m above the surface.

To reflect the above in our manuscript, the text now reads:

*Section 3.4: "Note that, in applying this 3D migration procedure, we inherently assume a uniform antenna radiation pattern. Although we acknowledge that this is not entirely correct for an unshielded dipole antenna, we see little indication in our data of a strong radiation pattern footprint. Indeed, diffraction circles in time slices show little variation in intensity*

*between the in-line and cross-line directions, and diffraction hyperboloids present in the unmigrated data are observed to correctly focus upon migration."*

Fig 2 and Table 1: I might be wrong, but I think photogrammetry was never mentioned before (or after) in the text. Even if the acquisition of the orthophotos was carried out by ETH Zürich's VAW Glaciology group, you should mention in the main text how you use this dataset and provide some details about it.

You are totally right; this is a mistake from our side. Information regarding the acquisition of the digital orthophotos and digital elevation models has been added at the end of Section 2.2. It now reads:

*Section 2.2: "Digital OrthoPhoto (DOP) and Digital Elevation Model (DEM) data were acquired and provided by ETH Zürich's VAW Glaciology group using a DJI Phantom 4 RTK drone. The surveys were conducted at a height of around 70 m above the glacier surface, with ground control points (between 8 and 12 depending on the surveys) randomly scattered across the area, for which precise locations were determined from a base station located close to the glacier. Overlapping was 70% forward, 60% sideways, and processing of the data was done using the Agisoft Metashape software."*

Please, when possible, substitute general statements with quantitative ones:

• L 118: "with a high level of repeatability for the horizontal positioning (Fig. 2e)." Can you quantify this?

Thank you for this comment. Please note that the horizontal positioning accuracy is already addressed in Section 3.2 with the statement:

*Section 3.2: "The mean distance between the center location of each bin and the true horizontal position of the GPR trace populating that bin was found to be 0.14m.".*

In Figure 2e, we show the acquisition maps superimposed over each other, as we feel that this is the best way to present the repeatability in Section 2.2, which deals with the data acquisition.

• L 119: "However, differences in vertical positioning between acquisitions were present due to glacier melting" Do you mean that the altitude above sea level has changed, but the altitude above the surface is always 5 m since it is controlled by the True Terrain Following? Can you quantify this change?

Thank you for this comment. Yes, the elevation above mean sea level changes because the drone system is set-up to fly at 5 m of height above the glacier surface which changes due to glacier melting (changes quantifiable as shown in Figure 2f). We have clarified the text in Section 2.2, which now reads:

*Section 2.2: "However, positioning differences in elevation a.m.s.l. between the acquisitions were present due to glacier melting, as the drone was programmed to fly at a height of 5 m above the ice surface (Fig. 2f). Note that the latter differences are accounted for in the depth imaging of the data (Section 3.4)."*

• L 120-122: "Advantages of a drone- based GPR acquisition are clear, as high- density data could not have been acquired on the glacier surface because of the large crevasses". What are these advantages? E.g., safety, time, difficulty in pulling a ground-based GPR over rough terrain...

Thank you for this comment. We revised this sentence which now reads:

*Section 2.2: "Advantages of a drone-based GPR acquisition in terms of efficiency, safety, and practicality are clear, as high-density data could not have been acquired on the glacier surface because of the large crevasses."*

• L 191-192: Considering the highly heterogeneous case study with ice, air and water, what is the associated error of using a single velocity? Can it be estimated, perhaps using bedrock depth from the four acquisitions?

While using a single velocity for ice is common when migrating glacier GPR data, we acknowledge that it introduces errors in the depth images due to the presence of water and air bodies within the ice. In our study, the only area where we expect significant error in the estimates of bedrock surface is beneath the air cavity. This is discussed in Section 3.5 and highlighted in Figures 7b-c. Note, however, that for determining the maximum cavity height (results provided in Table 2), we accounted for the presence of air within the cavity. These points are addressed in the following statements in the revised manuscript.

*Section 3.4: "Considering a constant radar velocity for glacier ice is a standard procedure for both ground-based and airborne GPR surveys (e.g., Langhammer et al., 2017; Grab et al., 2021; Church et al., 2020), even if the effects of internal heterogeneities like water- or air-filled features are neglected."*

*Section 3.5: "It is important to note that an apparent circular rise in this surface is observed at the location of the collapse feature. This is a migration artifact that results from the assumption of a constant radar wave speed in the glacier ice, when in fact an air-filled cavity exists at this location. However, this bedrock "pull-up" artifact does not impact the bedrock reflection amplitude analysis described below."*

*Section 3.5: "Regarding the maximum cavity height, note that they have been corrected for the presence of air within the cavity."*

**Results and Discussions**

L 274-275: The picking process for the air cavity should be introduced earlier in the methods.

Thank you for pointing this out. The picking process has been moved where it belongs to the Section 3.5.

Table 2: Would it be possible to provide an error for the measurements?

Unfortunately, we do not yet have a reliable method for quantifying the error associated with the inferred values for the minimum ice roof thickness and maximum cavity height. For this reason, we chose to not discuss the specific numerical values presented in Table 2 in the paper

text, but rather to focus on the trends observed in these numbers (see extracted sentences below from sections 4 and 5). While uncertainties certainly exist, stemming from factors such as the velocity model used, the manual picking process, and the binning of the data which slightly alters the trace locations, we cannot yet quantify them in a meaningful manner.

> *Section 4: "Tracking the evolution of the cavity over the summer of 2022, it appears that the ice roof becomes thinner while the height of the cavity increases (Table 2)."*

> *Section 5: "The results show that the thickness of the ice roof decreases while the height of the cavity increases as the collapse feature evolves throughout the summer (Table 2)."*

L 302-304 - "Regarding the two subglacial channels, the main one, originating from the northeast, is likely to drain the majority of the glacier's subglacial water system, whereas the second one, originating from the southeast, likely drains a constrained hydrological basin on the orographic left-hand side of the glacier." Could you provide some information to explain why you think this?

This interpretation is based on the knowledge about the glacier-wide topography of both the surface (DEMs available from the Swiss Federal Office of Topography) and the subglacial bedrock (Grab et al., 2021[1]), as well as our expert judgement, which is itself reliant on qualitative field observations. The latter, for example, entails the loudness of "water-gurgling sounds" that can be heard in the field during summer: although we never characterized these sounds in detail, they tend to be louder and have a lower-frequency tone on the northeast part of the glacier than on the southeast, suggesting – as written – that more water reaches the cavity from one side of the glacier than the other. To give some indications for this, we reworded the sentence as follows:

> *Section 5: "For what the relative magnitude of the two subglacial channels is concerned, the qualitative interpretation of glacier-wide DEMs available for both the glacier surface and subglacial topography (data from Swisstopo (2024) and Grab et al. (2021), respectively) as well as expert judgement based on in-situ perceptions (e.g. the "loudness" of water-related sounds that can be heard emerging from the glacier interior), suggest that the channel originating from the northeast is likely to ..."*

**Broader implications**

While the paper provides a thorough examination of a specific glacier collapse, it could enhance its impact by more explicitly contextualizing this phenomenon within the broader framework of global warming. Currently, the connection between the findings and global warming is only briefly mentioned through the reference (Egli et al., 2021b). While it is clear that a single collapse event cannot be directly attributed to the ongoing climate change, the increasing frequency of such events is linked to rising temperatures. Adding one or two sentences to address this point would help draw attention to the broader relevance of glacier snout collapses, which are not only indicative of cryospheric changes but can also have significant implications for human safety in mountain environments. This discussion could be incorporated into the Discussion or Conclusions sections, highlighting the importance of monitoring these phenomena in the context of climate-driven hazards.

We thank the reviewer for this important comment, highlighting the potential relevance of our work in the larger context. The reviewer is absolutely correct in saying that a link between the appearance of glacier collapse features and changes in climate cannot be established based on a single case study alone, but we take the opportunity to link our work to complementary work that has been performed in the meanwhile. We do this in the Conclusions, where we added the following paragraph:

> *Section 6: "On a broader perspective, and whilst acknowledging that an investigation focusing on an individual collapse feature cannot be used to establish a potential link to ongoing climate change, our study provides specific, complementary information to larger-scale studies that have provided evidence for an increase in the frequency by which surface collapse features occur (Egli et al., 2021b; Hösli et al., 2025). In combination, this growing body of literature sheds light on a phenomenon that has sparked curiosity in the recent past, and clearly associates it with the ongoing process of glacier thinning and related reduction in ice-flow velocities – two processes that are clearly driven by climate change and rising temperatures in particular (e.g., Hugonnet et al., 2021; Troilo et al., 2024; The GlaMBIE Team, 2025). As we expect that glacier surface-collapse features will emerge in other parts of the world too as glaciers continue to thin, our study contributes to better understand the local-scale processes and effects that such features have."*

**Technical comments:**

L 62-64: I suggest moving this paragraph after the discussion on terminal collapses (L75) to consolidate all relevant content in one section.

L 82: changes in à changes of

L93 tongue --> terminus

L 95: The reference to "boxes b-c" and "d-e" in Figure 1 could be clarified by separating these into distinct references for each sentence.

Fig 1: Consider making box (a) as wide as boxes (b+c) and highlighting the crevasses and collapse features in boxes (b), (c), and (d).

Thank you for these five suggestions which were addressed in the manuscript.

L 146-149: The first 3 sentences fit more into the acquisition section. I suggest moving them.

While it could certainly fit well in the acquisition section, we believe that keeping this information at the beginning of the data synchronization section provides a clearer narrative and enhances understanding.

L 189: Wasn't the height 5 m above the surface?

Indeed, the programmed height above the surface is set to 5 m. However, due to the flight velocity and the system's attempt to follow the glacier surface topography in real-time using the TTF system, the true height does vary throughout the survey. It is important to note that our

altimeter provides precise measurements (with 2 cm precision), and that these data are used for the processing of the results. To address this comment, we have added the following statement:

*Section 3.4: "Although the programmed drone flight height above the glacier surface was set to 5 m, this value varies during acquisition due to the flight velocity and the drone's attempt to follow the glacier surface topography in real-time using the TTF system."*

L231 sentences --> paragraph

L238 Please define DOP (it was defined in the label of Fig. 2, but should be defined also in the main text).

L 262-263: This sentence fits more in the methods section than in the results.

Fig 9.: "Elevation" in the y-axis label could be repeated only once per side.

Thank you for these last four suggestions which were addressed in the manuscript as well.

**Reviewer 2:**

We sincerely thank Reviewer 2 for their positive and detailed review, which helped us to significantly improve the completeness and depth of the manuscript. We are grateful for the attention they brought to the lack of clarity on some of the methodological aspects of our work (e.g., antenna specifications, acquisition details, and data processing). The question regarding sediment involvement and initial cavity formation helped strengthen the discussion by pushing us to contextualize our interpretations more carefully.

**Detailed comments**

L 83: Is this (2022) the latest GLAMOS reference for Rhone?

Thank you for pointing this out. All information was updated according to the latest GLAMOS report (2024), and the reference list was modified accordingly.

L 92: word order: "...investigated by Church et al. (...) **using GPR to**..."

Thank you for this comment. We have modified the sentence which now reads:

*Section 2.1: "The lower ablation zone of the glacier was previously investigated by Church et al. (2019, 2020, 2021) with 2D and 3D GPR using 25-MHz antennas to characterize and monitor the englacial and subglacial drainage network."*

Figure 1, L 99: which type of satellite image / source?

Thank you for this remark as there was a mistake: the background image is not from satellite imagery but from the orthophoto mosaic SWISSIMAGE 10 cm. The accurate reference was added to the list, with the following link providing all the related information: https://www.swisstopo.admin.ch/fr/orthophotos-swissimage-10-cm. The caption now reads:

*Figure 1: "Inset image from the Swiss Federal Office of Topography (Swisstopo, 2024), and background orthophotos from SWISSIMAGE 10 cm (Swissimage, 2024)."*

L 111: Impressive antenna. What is its weight? ("featherweight")
L 112: "transmitter-receiver"?

Thank you for the positive comment on the antenna. It's weight is 250g. It was made by applying copper tape and resistors to a piece of foam pipe insulation. The corresponding information can be found in Ruols et al. (2023). In the current publication, we modified this sentence which now reads:

*Section 2.2: "...(iv) a self-developed, ~80-MHz center-frequency, lightweight (250-g), unshielded, resistively loaded dipole antenna acting as both transmitter and receiver."*

L 116: Did you conduct tests for along-glacier-flow direction? (asking out of curiosity)

Along-glacier-flow data were indeed acquired with the drone-based GPR system on the Findelen glacier (Switzerland) in the summer of 2023. These data are currently being analyzed

with the objective of investigating internal deformation regarding ice dynamics. However, none were acquired over the Rhône glacier collapse feature.

L 119: Mostly out of curiosity: Would it have been useful to try and maintain a similar flying height as the previous flight despite glacier melting (e.g., changing the height above the ice, or using the flying heights of previous flights)? Or would that change the signal too much as the distance between antenna and ice increases, and the coupling to the ice surface therefore changes?

Thank you for this question. We have not tested this yet as, until now, we have relied upon the SPH True Terrain Following system to navigate along the glacier surface at a prescribed height of 5 m. Future work will involve navigation based on a recently acquired DEM, which should help to render the acquisition surface smoother. Regarding antenna coupling, at a height of 5 m above the ice, we are effectively coupled to air and not influenced by the ice. Finally, we have recently developed an efficient post-stack reverse-time-migration code for imaging our glacier datasets, which enables us to obtain accurate depth images independent of variabilities in the acquisition surface.

Figure 2: (e) impressive positioning precision between different dates. (f) Maybe name the y-axis "acquisition elevation" for clarity?

Thank you for the positive comment and suggestion. To keep consistency between figures in the manuscript, we prefer to keep the y-axis named "Elevation a.m.s.l. [m]". We hope that this is clear enough for the reviewer.

L 144: Maybe elaborate a bit more, in 1-2 additional sentences?

Thank you for this suggestion, the text now reads:

*Section 3: "Our data processing workflow transforms the acquired raw GPR measurements into a 3D reflection data volume, imaged in depth, which we use to explore the internal structure of the Rhône subglacial cavity and drainage channels. This workflow involves: (i) synchronization of the drone navigation and GPR data, (ii) binning of the consecutive 3D GPR datasets onto a common grid, (iii) creation of 3D data volumes followed by basic trace processing, (iv) 3D migration, and (v) modeling of the air cavity shape along with bedrock amplitude analysis."*

L 151: Was the same recording frequency used as in Ruols et al. (2023)?

Yes, the drone-based GPR system, including its recording frequency, is the same as in Ruols et al. (2023). This is highlighted in the text:

*Section 2.2: The datasets were collected using the recently developed drone-based GPR system of Ruols et al. (2023).*

L 152: You might want to explain that several flights were needed to change batteries. Knowing that this is explained in Ruols et al. (2023) as well.

Thank you for this comment. After much deliberation, we have decided to delete this sentence. As the data are segmented into profiles (as explained in Section 3.2), we believe that introducing this sentence makes the understanding unclear for no good reason.

L 160: What was the GPR recording frequency?

The GPR antenna center frequency was ~80 MHz.

*Section 2.2: "...(iv) a self-developed, ~80-MHz center-frequency, lightweight (250-g), unshielded, resistively loaded dipole antenna acting as both transmitter and receiver."*

L 190: You might want to provide 2-3 specific original references justifying the chosen velocity of 0.167 m ns-1.

Thank you for this suggestion. The text now reads:

*Section 3.4: "...(ii) a lower ice layer with velocity 0.167 m ns-1 (e.g., Murray et al., 2000; Church et al., 2021; Egli et al., 2021a).".*

Figure 4 / L207: "..325 ns (purple)" : I see this as blue.

Thank you for this comment which also was pointed out by some of the co-authors. This was a mistake while creating the figure, and the purple color of the depth slices shown in Figure 4 is now the same than for the depth slices from Figure 6.

Figure 5: It might help to add a legend for the blue, yellow, red arrows in the figure.

Thank you for this suggestion, which was addressed in the revised version of Figure 5.

Figure 6: What features or situation can we see in the depth slices g, h, i?

Thank you for this question. A paragraph was added to the section 4. To keep consistency with other figures, arrows and legend were added to Figure 6, which makes the description easier.

*"Section 4: The 3D visualization of the July 2022 dataset presented in Figure 6 clearly shows a number of glacier internal structures. In the inline and crossline profiles in Figures 6a-f, strong reflections related to the bedrock interface (yellow arrows), the air cavity roof (blue arrows), and dipping crevasses (black arrow) can be observed. In the depth slice presented in Figure 6h, we see large circular reflections from dipping crevasses (black arrows) which can be viewed in greater detail and context in video supplement V5. The depth slice in Figure 6i reveals the walls of the circular air cavity (blue arrows), and a suspected subglacial channel leaving the collapse feature (red arrow), which is also clearly seen in the results of our amplitude analysis discussed below. For further visualizations, videos V3 to V14 contain animations providing further insights into the 3D models obtained for all four surveys."*

L 221: Remove "Indeed,"

Thank you, the modification is done.

L 234: Could this consideration of maximum reflection strength over a 2-m window introduce some sort of bias or artefact?

Tests conducted in the context of this work as well as for previous works suggest that taking the maximum instantaneous amplitude value over this 2-m-wide window, approximately equal to the dominant GPR wavelength in ice, provides the best results when it comes to quantifying the bedrock reflection strength and identifying the presence of subglacial channels. Although use of the maximum rather than mean results in a slightly noisier image, we have found that it offers heightened sensitivity to the presence of channels.

L 238: Did you ever introduce "DOP" (Digital Orthophoto)?

Thank you for pointing this out as well. The digital orthophotos and digital elevation models are now introduced in the data acquisition section:

*Section 2.2: "Digital OrthoPhoto (DOP) and Digital Elevation Model (DEM) data were acquired and provided by ETH Zürich's VAW Glaciology group using a DJI Phantom 4 RTK drone. The surveys were conducted at a height of around 70 m above the glacier surface, with ground control points (between 8 and 12 depending on the surveys) randomly scattered across the area, for which precise locations were determined from a base station located close to the glacier. Overlapping was 70% forward, 60% sideways, and processing of the data was done using the Agisoft Metashape software."*

Figure 8: Maybe a detail, but still worth mentioning for future / further investigation: There is a strong high amplitude signal visible in the lower corner of each plot (25N / 130 E), maybe indicating the edge of another channel, or ponding. Alternatively, it could be an artefact, as it is on the edge of the dataset.

Thank you for noticing this. This artefact is now introduced in the text as follow:

*Section 4: "Note that there is another high amplitude anomaly, located in the lower right corner of each subfigure, which may represent another water body or cavity, but lack of additional data prevents us from drawing any further conclusions."*

Also figure 8, L 252: "..leaving the feature **westwards**.."

Done.

L 273: "....due to a combination of ice creep into the cavity and partial mechanical failure." Maybe be a bit more careful with this statement and present it as a hypothesis?

Thank you. The text now reads:

*Section 4: "This is a result of both surface ice melt and subsidence, the latter possibly being due to a combination of ice creep into the cavity and partial mechanical failure".*

L 310: "..**over** time"

L 321: "...evolves **throughout** summer "

Thank you, these two modifications are done.

L 328: Mechanical failure (and erosion of subglacial till) was, among others, also hypothesized by Egli et al. (2021b), but under the name of "block caving" (Paige, R. (1956). Subglacial stoping or block caving: A type of glacier ablation. *Journal of Glaciology*, **2**, 727–729. **https://doi.org/10.3189/s0022143000024977**). Very similarly to Rhône, ice blocks floating out of the terminus at Otemma were observed already in summer 2017 – the year before the ice surface collapse event. But no borehole was made to verify if a cavity had started to form while the glacier outlet channel was still pressurized. The correct main finding for Rhône remains that the outlet channel at Rhône seems to have remained pressurized for several weeks while a large cavity was forming underneath the ice.

We thank the reviewer for this comment and the description of additional observations at both Otemma and Rhone. As we understand these observations being unpublished, we do not know how we would be able to mention them in our revised text. We therefore only extended the original sentence as to refer to the works of Egli et al. (2021b) and Paige (1956).

> *Section 5: "As an alternative, Räss et al. (2023) hypothesized that the collapse feature at Rhône glacier grew by mechanical failure of ice lamellas and the subsequent melting and transport of the ice by the subglacial stream – a process referred to as "block caving" in Paige (1956) or Egli et al. (2021b)."*

L 333: This is an interesting and valuable discussion. You could talk a bit more about other potential mechanisms for channel widening and cavity opening, namely sediment erosion (and deposition). Did you determine whether the ground below the collapse feature is / was mainly composed of bedrock, or also sediments? Or, if there used to be sediments, but they were eroded away by the subglacial channel during the formation of the collapse feature?

This also raises the question about what initiated the formation of the first cavity, making flow non-pressurized, and which then led to roof destabilization, detachment of lamellae, etc., to start with?

These are all important questions to which we would very much like being able to answer. Unfortunately, we cannot, as our information is constrained to the data we collected during the GPR survey and to visual observations that we gathered while being in the field. What we can say is that there were sediments visible at the base of the cavity once it collapsed (cf. Figure 11) but that we are unable to make any statement about the thickness of these sediments and, even less, about their role in the cavity-formation process (we now better discuss this potential role in reply to the reviewer's question below). As for the initiation of the first cavity, we presently subscribe to the hypothesis that this was linked to the thermo-mechanical erosion of a part of the ice-channel walls facilitated by the water turbulence caused by the bedrock step that is visible after the cavity collapse (see white arrow in Figure 11). This hypothesis is presented in the section 5 and now reads as follow. We remain open for concrete suggestions on how to potentially expand our line of argumentation in this respect.

> *Section 5: "The results of the amplitude analysis (Fig. 8) suggest that the Rhône collapse feature formed at a particular location near the glacier snout where (i) the main subglacial channel forms a meander, and (ii) a secondary channel enters the main channel. Examination of a photograph taken in the summer of 2023 after full collapse of the cavity (Fig. 11) also indicates a localized step (~1 m in height) in the bedrock topography, which was further confirmed in Hösli et al. (2025). The combination of these three factors could*

*have led to water turbulence and related energy dissipation, which could ultimately be the process by which the cavity was initiated."*

Figure 11: There are lots of (partially eroded) sediments, and bedrock, visible in this picture. You should talk about this in the discussion, and about the sediments' potential role in the initial formation of the cavity.

We thank the reviewer for this comment. Indeed, the possibility that sediments might play a role in the formation of subglacial cavities, was prominently suggested by Stocker-Waldhuber et al. (2017) who observed accelerated ice-surface subsidence for an Austrian glacier after the evacuation of a substantial amount of subglacial sediments triggered by a heavy precipitation event. While we do not have any evidence for a similar process to have played a role in the case of Rhône glacier, we now explicitly discuss the above possibility which now reads:

*Section 5: "In contrast, although sediments are visible at both the glacier terminus and inside of the collapsed feature (Figure 11), we do not have any evidence that would suggest that these sediments played a significant role during the formation of the feature itself. This is dissimilar to the situation described by Stocker-Waldhuber et al. (2017), who analyzed the formation of a surface collapse feature for Gepatschferner, Austria, and reported that the subsidence of the glacier surface accelerated significantly after a heavy precipitation event evacuated a sediment layer of more than 10 m thickness."*

---

## Author Response (AR2)

**To editor Kristin Poinar:**

We sincerely thank Prof. Kristin Poinar once again for her thoughtful and constructive feedback, which significantly improved the overall quality of the manuscript. We also appreciate her availability and excellent management of the review process, which greatly facilitated our efforts in revising the manuscript. Please find below the modifications made regarding the final revision.

Line 90: "precise drone positioning" In keeping with a request from Reviewer 1, please add quantitative information here. In the response document, you mentioned RTK maintained the drone altitude at ± 2 cm accuracy; please add this here.

Thanks for the comment, it now reads:

*"The use of Real Time Kinematic (RTK) technology enables navigational positioning accuracy of approximately 1 cm horizontally and 2 cm vertically, and allows for the accurate repetition of flight paths, making high-density 4D GPR acquisitions over inaccessible terrains and further 4D glaciological investigations feasible."*

Line 485: Similar comment, why not replace "over an approximately one-year period" with a more precise measure of time, e.g. 14-16 months or whatever your true findings are.

Thanks for the comment, it now reads:

*"In this paper, we present a high-density, high-resolution 4D GPR dataset acquired over the same surface collapse feature investigated by Hösli et al. (2025) on the Rhône glacier, which initially developed near the glacier terminus in late 2021, evolving over 20 months and collapsed in June 2023."*

Line 687: When describing the survey design, I would find it helpful if you mentioned the wavelength of the radar center-frequency in ice, since it relates to the survey line spacing design. By my calculation it was 2.1 m, but a reader should not have to calculate it on their own.

Thanks for the comment, it now reads:

*"The survey trajectories were planned with the Universal ground Control Software (UgCS), with a survey line spacing of 1 m (the wavelength of the antenna center-frequency in ice is ~2.1 m..), a target height above the glacier surface of 5 m, and a flight speed of 4 m s-1 (Ruols et al., 2023)."*

Lines 917, 928, 930: Change "what is done" and "is also modeled" and "is followed by" and "is computed for" to active voice. Most of this section is correctly in active voice, but these phrases are in passive voice and do not match.

Thank you, the entire paragraph was addressed.

Figure 7 caption: Explain the black lines (contours) and state their contour interval.

Thank you, the caption now reads:

*"[…] (c) 3D view of the final bedrock surface model, with black lines indicating elevation isolines at 2-m intervals."*

Line 1008: "the lower right corner of each subfigure" I suggest to change this to "the southernmost corner of the surveyed area" in order to refer to the actual place and direction, rather than a mapped representation of it.

Thank you, the modification was done.

Line 1350: "For what the relative magnitude of the two subglacial channels is concerned" - some grammatical / word parsing issues here.

Line 1351-2: Cite the Swisstopo and Grab et al. in turn, i.e. "glacier-wide DEMs for the glacier surface (Swisstopo, 2024) and subglacial topography (Grab et al., 2021) as well as expert..."

Thank you for the two comments, it now reads:

*"Considering (i) the relative magnitude of the two subglacial channels, (ii) the qualitative interpretation of glacier-wide DEMs for the glacier surface (Swisstopo, 2024) and subglacial topography (Grab et al., 2021), and (iii) expert judgment based on in-situ perceptions (e.g., the 'loudness' of water-related sounds emerging from within the glacier), it seems that the channel originating from the northeast likely drains the majority of the glacier's subglacial water system, whereas the second channel, originating from the southeast, appears to drain a constrained hydrological basin on the orographic left-hand side of the glacier."*

Figure 13 caption: Name the date that these stills were acquired.

Thank you, the modification was done.

Line 1457: "On a broader perspective," - remove this phrase, as conclusions are always taken from a broad perspective.

Thank you, the modification was done.